## Report

# Myc-dependent endothelial proliferation is controlled by phosphotyrosine 1212 in VEGF receptor-2

Chiara Testini[1,†], Ross O Smith[1], Yi Jin[1], Pernilla Martinsson[1], Ying Sun[1], Marie Hedlund[1], Miguel Sáinz-Jaspeado[1], Masabumi Shibuya[2], Mats Hellström[1] & Lena Claesson-Welsh[1,*]

## Abstract

Exaggerated signaling by vascular endothelial growth factor (VEGF)-A and its receptor, VEGFR2, in pathologies results in poor vessel function. Still, pharmacological suppression of VEGFA/VEGFR2 may aggravate disease. Delineating VEGFR2 signaling *in vivo* provides strategies for suppression of specific VEGFR2-induced pathways. Three VEGFR2 tyrosine residues (Y949, Y1212, and Y1173) induce downstream signaling. Here, we show that knock-in of phenylalanine to create VEGFR2 Y1212F in C57Bl/6 and FVB mouse strains leads to loss of growth factor receptor-bound protein 2- and phosphoinositide 3′-kinase (PI3K)p85 signaling. C57Bl/6 *Vegfr2*[Y1212F/Y1212F] show reduced embryonic endothelial cell (EC) proliferation and partial lethality. FVB *Vegfr2*[Y1212F/Y1212F] show reduced postnatal EC proliferation. Reduced EC proliferation in *Vegfr2*[Y1212F/Y1212F] explants is rescued by c-Myc overexpression. We conclude that VEGFR2 Y1212 signaling induces activation of extra-cellular-signal-regulated kinase (ERK)1/2 and Akt pathways required for c-Myc-dependent gene regulation, endothelial proliferation, and vessel stability.

**Keywords** angiogenesis; GRB2; Nck; PI3Kp85; proliferation

**Subject Categories** Development & Differentiation; Signal Transduction; Vascular Biology & Angiogenesis

## Introduction

Vascular endothelial growth factor (VEGF)-A is a potent pro-angiogenic factor regulating endothelial cell (EC) proliferation, migration, vessel lumen formation, vessel pruning, and stability [1,2]. VEGFA binds to VEGF receptor (VEGFR)-1 (also denoted Flt1) and VEGFR2 (denoted KDR in the human and Flk1 in the mouse) [3]. VEGFR2 is known to be a critical regulator of endothelial biology.

Binding of VEGFA to VEGFR2 leads to activation of the receptor tyrosine kinase and phosphorylation of tyrosines (Y) 949, 1052, 1057, 1173, and 1212 (mouse VEGFR2 sequence numbering) in the receptor intracellular domain [4]. Phosphorylated (p) Y949 in VEGFR2 mediates c-Src activation at EC junctions, and loss of pY949 signaling in *Vegfr2*[Y949F/Y949F] mice results in suppressed vascular leakage [4–7]. Phosphorylation of Y1052 and Y1057, located on the kinase domain activation loop, is required for full induction of VEGFR2 kinase activity [8]. pY1173 binds phospholipase C-γ (PLCγ) [9,10] and the Src homology-2 (SH2) adapter proteins SHB and SHC [11,12]. pY1173 downstream signaling induces the extracellular-signal-regulated kinase (ERK) 1/2 pathway, promoting EC proliferation [10,13]. SHB-dependent signaling has been implicated in EC attachment and migration [14]. *Vegfr2*[Y1173F/Y1173F] mice die at embryonic day 8.5 (E8.5), due to suppressed EC differentiation, recapitulating the phenotype of the global *Vegfr2/flk1* knockout mouse [15,16].

Less is known about the downstream signal transduction initiated by pY1212 in VEGFR2. Based on *in vitro* studies, pY1212 presents a binding site for the adaptor Nck, allowing recruitment of the tyrosine kinase Fyn [17] and heat-shock protein 27-mediated actin polymerization and EC migration [17,18]. pY1212 has also been implicated in activation of the Roundabout (Robo) receptor by its ligand Slit. In this model, Robo and VEGFR2 are both required for binding Nck, leading to Cdc42 activation and EC front–rear polarity [19].

Here, we have investigated the *in vivo* role of pY1212 in VEGFR2 signaling using a Y1212F knock-in mouse model, *Vegfr2*[Y1212F/Y1212F] [15], on C57Bl/6 and FVB mouse backgrounds. pY1212 creates a binding site for growth factor receptor-bound protein 2 (GRB2) and the regulatory p85 subunit of phosphoinositide 3′-kinase (PI3Kp85), mediating activation of ERK1/2 and Akt pathways, respectively. ERK1/2 and Akt pathways positively and negatively regulate the expression of c-Myc [20,21], a transcription factor that controls expression of a broad set of genes involved in cell cycle progression, cell proliferation, apoptosis, and cell metabolism [20,22]. We show that pY1212 is required for VEGFR2-dependent EC proliferation in a c-Myc-dependent manner and that the temporal establishment of the

1   Department of Immunology, Genetics and Pathology, Rudbeck Laboratory, Science for Life Laboratory, Uppsala University, Uppsala, Sweden
2   Institute of Physiology and Medicine, Jobu University, Takasaki, Gunma, Japan
    *Corresponding author. Tel: +46 7016 79260; E-mail: lena.welsh@igp.uu.se
    †Present address: Transplant Research Program, Boston Children's Hospital, and Harvard Medical School, Boston, MA, USA

$Vegfr2^{Y1212F/Y1212F}$ vascular phenotype differs between C57Bl/6 and FVB mouse strains.

# Results

## Partial embryonic lethality of $Vegfr2^{Y1212F/Y1212F}$ on the C57Bl/6 background

A point mutation was introduced in exon 27 of the *Flk1* gene (Fig EV1A and B) to replace tyrosine 1212 with phenylalanine, creating $Vegfr2^{Y1212F/Y1212F}$ (previously denoted Flk-1$^{1212F/1212F}$ [15]). While originally reported to lack a phenotype, extensive back-crossing of the $Vegfr2^{Y1212F/Y1212F}$ mutation onto the C57Bl/6 background was accompanied by partial embryonic lethality. At E11.5, the $Vegfr2^{Y1212F/Y1212F}$ embryos appeared at a 25% Mendelian frequency (29/114; $Vegfr2^{Y1212F/Y1212F}$/all E11.5 embryos). At E14.5, 19% (13/70) were of the mutant genotype, while at birth, only 13% (40/314) were mutants (Fig 1A). Mutant embryos succumbed without signs of hemorrhage or edema (Fig EV2A). Analysis of the embryonic vascular development showed reduced vessel density and suppressed EdU incorporation indicating reduced EC proliferation, in $Vegfr2^{Y1212F/Y1212F}$ E11.5 hindbrains compared to WT (Fig 1B–D). The decrease in overall vessel density occurred without changes in EC density within the vessels, i.e., EC morphology was unaffected by the mutation (Fig EV2B). Vascular density and EC proliferation in the E11.5 mutant yolk sacs were unchanged (Fig EV2C and D), as was the proportion of CD31$^+$/CD45$^-$ ECs in the E11.5 mutant compared to WT embryos, as determined by flow cytometry (Fig EV2E and F). Likewise, cardiac function and morphology appeared normal when examining embryonic hearts (Fig EV2G–I). These data suggest that while vascular development was affected in the hindbrain, the $Vegfr2^{Y1212F/Y1212F}$ C57Bl/6 mutant embryos did not show signs of a major developmental insult. Instead, the $Vegfr2^{Y1212F/Y1212F}$ genotype conferred an EC dysfunction that led to gradual death resulting in loss of about 50% of the mutant embryos *in utero*.

Surviving $Vegfr2^{Y1212F/Y1212F}$ mice developed normally although, starting from week 11, they weighed significantly less than WT mice (Fig 1E). Postnatal C57Bl/6 vascular development in the P6 $Vegfr2^{Y1212F/Y1212F}$ retina [23] was, however, unaffected with regard to overall retinal vessel outgrowth, vascularized area, total EC, and vessel density as well as EC proliferation, compared to wild-type (WT) littermates (Figs 1F–J and EV2J). Also, blood cell parameters were unaffected by the $Vegfr2^{Y1212F/Y1212F}$ mutation indicating that the hematopoietic lineage developed normally (Table 1).

## Impaired $Vegfr2^{Y1212F/Y1212F}$ postnatal vascular development on the FVB background

The $Vegfr2^{Y1212F/Y1212F}$ mutation was backcrossed onto the FVB strain. FVB mutant pups were born at the normal 25% Mendelian ratio (89/336; $Vegfr2^{Y1212F/Y1212F}$/all), and hematopoiesis was unaffected (Table 1). The weight development of $Vegfr2^{Y1212F/Y1212F}$ was comparable to that of WT pups (Fig 2A).

FVB $Vegfr2^{Y1212F/Y1212F}$ E11.5 hindbrains displayed unperturbed EC proliferation and vessel density (Figs 2B and C, and EV3A and B).

On the other hand, retinal vascular development in $Vegfr2^{Y1212F/Y1212F}$ P4–P7 FVB pups showed a significantly delayed radial outgrowth and a reduced vascularized area compared to WT (Fig 2D; quantifications in Fig 2E and F). Moreover, although vessel density was overall normal (Fig EV3C), the density in the vascular front (outer 20% vascularized area; pink in Fig 2D) was significantly reduced in the $Vegfr2^{Y1212F/Y1212F}$ P6 and P7 pups (Fig 2G). EdU incorporation was reduced in $Vegfr2^{Y1212F/Y1212F}$/FVB P6 retina vascular front compared to WT littermates (Fig 2H and I), contributing to the reduced vascular density in the mutant pups. Sprouts along the vascular front in $Vegfr2^{Y1212F/Y1212F}$ P6 retinas were longer than in WT pups (Fig 2D inset; panel J), indicating disturbed remodeling of the developing vasculature. The abundance of filopodia extending from the sprouting tip cells was, however, normal (Fig EV3D and E).

Decreased proliferation may affect remodeling and vessel stability, leading to increased pruning in the retina [24]. Collagen IV basement membrane sleeves devoid of endothelial cells (empty sleeves) constitute a vessel instability marker [25]. The number of empty sleeves was significantly increased in the $Vegfr2^{Y1212F/Y1212F}$/FVB P4 retina (Figs 2K and L, and EV3F), after which it returned to WT levels. Thus, the $Vegfr2^{Y1212F/Y1212F}$ FVB postnatal retinal vasculature showed reduced EC proliferation and vessel stability, both contributing to the reduced vessel density.

The retinal vasculature is protected by the retina–blood barrier with unique properties compared to other vascular beds. Therefore, we also examined the tracheal vasculature, which becomes established through angiogenic sprouting in a VEGFA-dependent manner [26]. The number of capillaries spanning tracheal cartilage rings in P1, P3, and P5 $Vegfr2^{Y1212F/Y1212F}$ pups was decreased compared to WT tracheas (Fig EV3G and H), and the number of collagen IV empty sleeves was increased in the P3 mutant trachea (Fig EV3I and J). Thus, although kinetically distinct from the findings in the retina, the pattern of decreased vascular density and increased regression was established also in the $Vegfr2^{Y1212F/Y1212F}$ trachea vasculature.

In contrast, EC apoptosis, front–rear polarity, migration, and lumen formation were comparable in $Vegfr2^{Y1212F/Y1212F}$ and WT pups (Fig EV4). Levels of cleaved caspase-3 in ECs proximal to collagen IV empty sleeves in P4–P6 pups were similar between $Vegfr2^{Y1212F/Y1212F}$ and WT littermate retinas (Fig EV4A and B), indicating that the mutation did not affect EC apoptosis [27,28]. EC front–rear polarity in the retinal vasculature, based on the location of the Golgi relative to the nucleus and the direction of outgrowth [29], was also unaffected in the mutant (Fig EV4C and D). Similarly, no polarity phenotype was detected in the tracheal capillaries at P3 (Fig EV4E and F). Wound healing analyses showed no difference in VEGFA-induced migration of ECs isolated from $Vegfr2^{Y1212F/Y1212F}$ or WT lungs (Fig EV4G). Lumen formation, assessed by immunostaining for the lumen marker podocalyxin [27] in the P3 tracheal vasculature, was also unaffected (Fig EV4H and I).

## GRB2 and PI3Kp85 bind to pY1212

To verify that the Y1212F knock-in did not affect the phosphorylation and therefore signaling through other VEGFR2

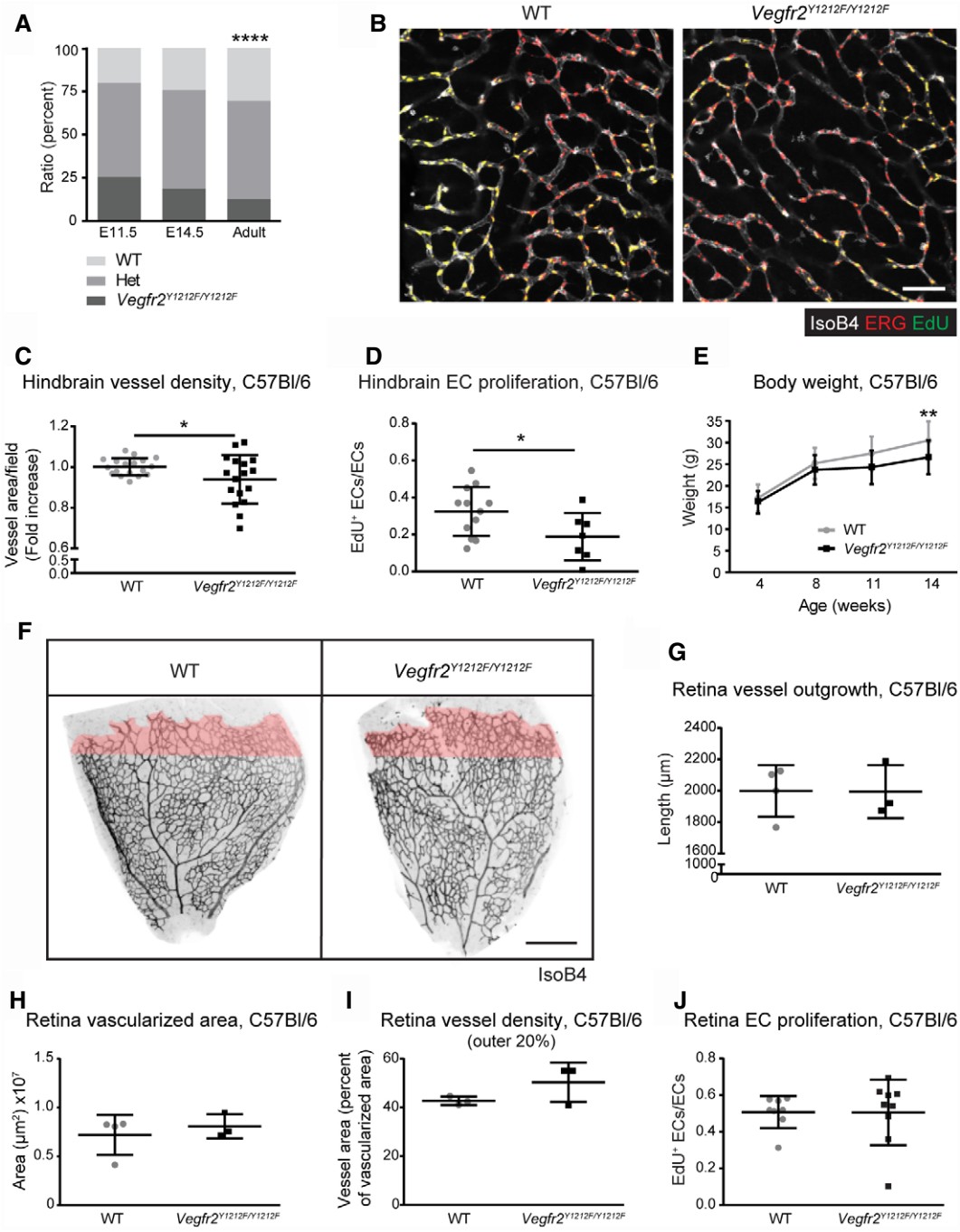

**Figure 1. Embryonic and postnatal vascular development in *Vegfr2^{Y1212F/Y1212F}* C57Bl/6 mice.**

A   Partial embryonic lethality after E11.5. Chi-square, ****$P < 0.0001$. E11.5 $n = 114$ embryos, E14.5 $n = 70$ embryos, adult $n = 314$ born pups.

B   Hindbrain vessel morphology. Isolectin B4 (IsoB4; white), ERG (red), and EdU incorporation (green). EdU signal was masked with ETS-related gene (ERG) signal. Scale bar, 50 μm.

C   Hindbrain vessel density. Isolectin B4 vessel area normalized to wild-type (WT) mean; one dot is two fields/mouse; error bars: SD; unpaired *t*-test, *$P < 0.05$. $n = 18$–20.

D   Hindbrain EC proliferation. EdU incorporation in ERG-positive ECs (double-positive cells, yellow in B). One dot is two fields/mouse; error bars: SD; unpaired *t*-test, *$P < 0.05$. $n = 7$–12.

E   Reduced weight gain in the adult. Error bars: SD; 2-way ANOVA $P = 0.0009$; Sidak's multiple comparison test, **$P < 0.01$. $n = 13$–20 mice.

F   P6 retina vessel morphology. Isolectin B4 (IsoB4) in black; the outer 20% of the vascularized area in pink. Scale bar, 250 μm.

G–J   Vessel parameters in the P6 developing retina. (G) Radial outgrowth (distance between optic nerve and vascular front). (H) Vascularized area. (I) Vessel area in the outer 20% of the retinal vasculature; pink in (F) (vessel area normalized to vascularized area). Each dot represents one retina/mouse; $n = 3$–4. (J) EdU incorporation in ERG-positive ECs normalized to total ERG-positive ECs. Each dot represents one retina/mouse; error bars: SD. $n = 8$–9.

**Table 1. Blood component count in wild-type (WT) and mutant ($Vegfr2^{Y1212F/Y1212F}$) C57Bl/6 and FVB mice.**

| | C57Bl/6 | | | FVB | | |
|---|---|---|---|---|---|---|
| | WT ($n = 3$) | $Vegfr2^{Y1212F/Y1212F}$ ($n = 3$) | *P*-value* | WT ($n = 3$) | $Vegfr2^{Y1212F/Y1212F}$ ($n = 3$) | *P*-value* |
| Red blood cells (×10^12/l) | 6.793 (±0.625) | 6.74 (±0.71) | 0.96 | 7.727 (±0.196) | 7.613 (±0.160) | 0.68 |
| White blood cells (×10^9/l) | 3 (±0.404) | 3.567 (±0.418) | 0.38 | 3.733 (±0.418) | 4.3 (±0.173) | 0.28 |
| Lymphocyte count (×10^9/l) | 2.633 (±0.338) | 3.4 (±0.5) | 0.24 | 3.2 (±0.351) | 3.5 (±0.577) | 0.45 |
| Platelet (×10^9/l) | 741.7 (±73.77) | 762 (±137.1) | 0.90 | 1,119 (±48.91) | 1,155 (±40.95) | 0.60 |
| Mean platelet volume (fl) | 6.067 (±0.088) | 5.867 (±0.033) | 0.10 | 6.9 (±0.208) | 6.733 (±0.120) | 0.53 |
| Hemoglobin (g/l) | 103.7 (±9.939) | 107.3 (±10.84) | 0.82 | 123.7 (±2.906) | 121.7 (±2.906) | 0.65 |

*Unpaired *t*-test.

phosphorylation sites, the degree of phosphorylation on Y949 and Y1173 was examined. Tail-vein injection of VEGFA induced strong phosphorylation of VEGFR2 at Y949 and Y1173 both in WT and in $Vegfr2^{Y1212F/Y1212F}$ FVB lung lysates at 1 min postinjection, while VEGFR2 Y1212 was phosphorylated only in WT mice (Fig 3A and B).

Phosphorylated tyrosines in receptor tyrosine kinases (RTKs) serve as specific binding sites for SH2 domain proteins [30,31]. To investigate pY1212 binding partners, we immobilized synthetic peptides covering the VEGFR2 Y1212 region, or as a control, the VEGFR2 Y949 region in VEGFR2, phosphorylated or not on Y1212 or Y949. Immobilized peptides were incubated with human dermal microvascular EC (HDMEC) lysate. Mass spectrometry analysis identified 10 proteins specifically binding to the pY1212 peptide, as judged by the number of tryptic peptides detected from each unique protein (PSM score), and to the peptide area score (see Materials and Methods), and the lack of specific enrichment to the unphosphorylated Y1212 matrix (Fig 3C). Note that Nck-derived peptides were not enriched by the pY1212 peptide [17]. Of the 10 specifically enriched proteins, GRB2, PI3Kp85, and tensin 1 (TNS1) are highly expressed in ECs. We excluded adrenomedullin (ADM) and GRB2-related adapter protein (GRAP) from further analyses due to their low expression in brain and lung microvascular ECs; see public database [32]. Moreover, ubiquitin-associated and SH3

domain-containing protein B (UBASH3B), solute carrier family 9 (sodium/hydrogen exchanger) isoform 3 regulatory factor 1 (SLC9A3R1), WAS/WASL interacting protein family member 2 (WIPF2), and acyl-CoA oxidase 1, palmitoyl (ACOX) 1, and 3 were excluded from further analyses as they lack SH2 domain motifs and/or are localized in subcellular compartments (e.g., in the peroxisome in the case of ACOX1 and ACOX3), where they are unlikely to interact with VEGFR2. GRB2, PI3Kp85, and TNS1 were further analyzed for binding to pY1212 by immunoblotting of samples from HDMEC lysates incubated with immobilized pY1212 peptide (Fig 3D). As the VEGFR2/TNS1 interaction was not confirmed by immunoblotting on such peptide-enriched samples, TNS1 was excluded from further investigation. The pY1212-dependent binding of GRB2 and PI3Kp85 to VEGFR2 was validated using proximity ligation assay (PLA) on ECs isolated from FVB lungs (iECs). The specificity of VEGFR2 interaction with Grb2/p85 in this assay was validated by omitting primary antibodies (see Materials and Methods). Both VEGFR2/GRB2 (Fig 3E and F) and VEGFR2/PI3Kp85 (Fig 3E and G) complexes were induced by VEGFA stimulation of WT iECs but not mutant iECs. The presence of VEGFR2 complexes with GRB2 and PI3Kp85 before VEGFA stimulation could depend on endogenous production of VEGFA by the iECs.

The nuclear accumulation of downstream signal transducers for GRB2, namely pERK1/2 (Figs 3H and I, and EV5A), and for

**Figure 2. Embryonic and postnatal development in $Vegfr2^{Y1212F/Y1212F}$ FVB mice.**

A   Body weight. P, postnatal day; w, weeks. Error bars: SD. $n = 5$–12 mice.

B, C   Hindbrain vessel parameters. (B) Vessel density; isolectin B4 vessel area normalized to wild-type (WT) mean of the litter. (C) EdU incorporation in ERG-positive ECs normalized to total ERG-positive ECs. Each dot represents two fields/mouse; error bars: SD; unpaired *t*-test, not significant. $n = 5$–11.

D   P4-P7 retina vessel morphology. Isolectin B4 (IsoB4) in black; outer 20% of the vascularized area in pink. Scale bar, 250 μm. P6 insets; sprout morphology. Scale bar, 50 μm.

E   Radial outgrowth. Distance between optic nerve and vascular front. P, postnatal day; error bars: SD; 2-way ANOVA $P < 0.0001$; Bonferroni's multiple comparison test, *$P < 0.05$. P4 $n = 3$–5 retinas, one retina/mouse, P5 $n = 4$–7, P6 $n = 9$–9, P7 $n = 4$–5.

F   Retina vascularized area. P, postnatal day; error bars: SD; 2-way ANOVA $P = 0.0013$. P4 $n = 7$–9 retinas, one retina/mouse, P5 $n = 4$–9, P6 $n = 9$–9, P7 $n = 4$–5.

G   Retinal vessel density in the retinal front. Quantification of the vessel area in the outer 20% of the vasculature (pink in D) (vessel area normalized to vascularized area). P, postnatal day; error bars: SD; 2-way ANOVA $P = 0.0054$; Bonferroni's multiple comparison test, *$P < 0.05$, **$P < 0.01$. P4 $n = 3$–5 retinas, one retina/mouse, P5 $n = 4$–6, P6 $n = 3$–5, P7 $n = 4$–5.

H, I   Proliferation of retinal ECs. (H) EdU incorporation in ERG-positive ECs normalized to total ERG-positive ECs, at P6. (I) Isolectin B4 (IsoB4; white), ERG (red), EdU (green) and double-positive cells (yellow). Lower panel shows ERG/EdU merged staining alone with EdU signal masked by ERG signal. Scale bar, 50 μm; error bars: SD; unpaired *t*-test, *$P < 0.05$. $n = 5$–8 retinas, one retina/mouse.

J   Retina sprout length. Sprout length from tip to base in P6 retinas; error bars: SD; unpaired *t*-test, *$P < 0.05$; $n = 3$–5 retinas, one retina/mouse.

K, L   Vessel stability. (K) Isolectin B4 (IsoB4, green) and collagen IV (red) in P4 WT and $Vegfr2^{Y1212F/Y1212F}$ retinas. Arrows: collagen IV-positive empty sleeves. A: artery. Scale bar, 50 μm. (L) Collagen IV empty sleeves around main arteries in P4–P7 retinas, three fields/retina/mouse. Error bars: SD; 2-way ANOVA $P = 0.0580$; Bonferroni's multiple comparison test, **$P < 0.01$. P4 $n = 7$–9, P5 $n = 6$–12, P6 $n = 3$–5, P7 $n = 4$.

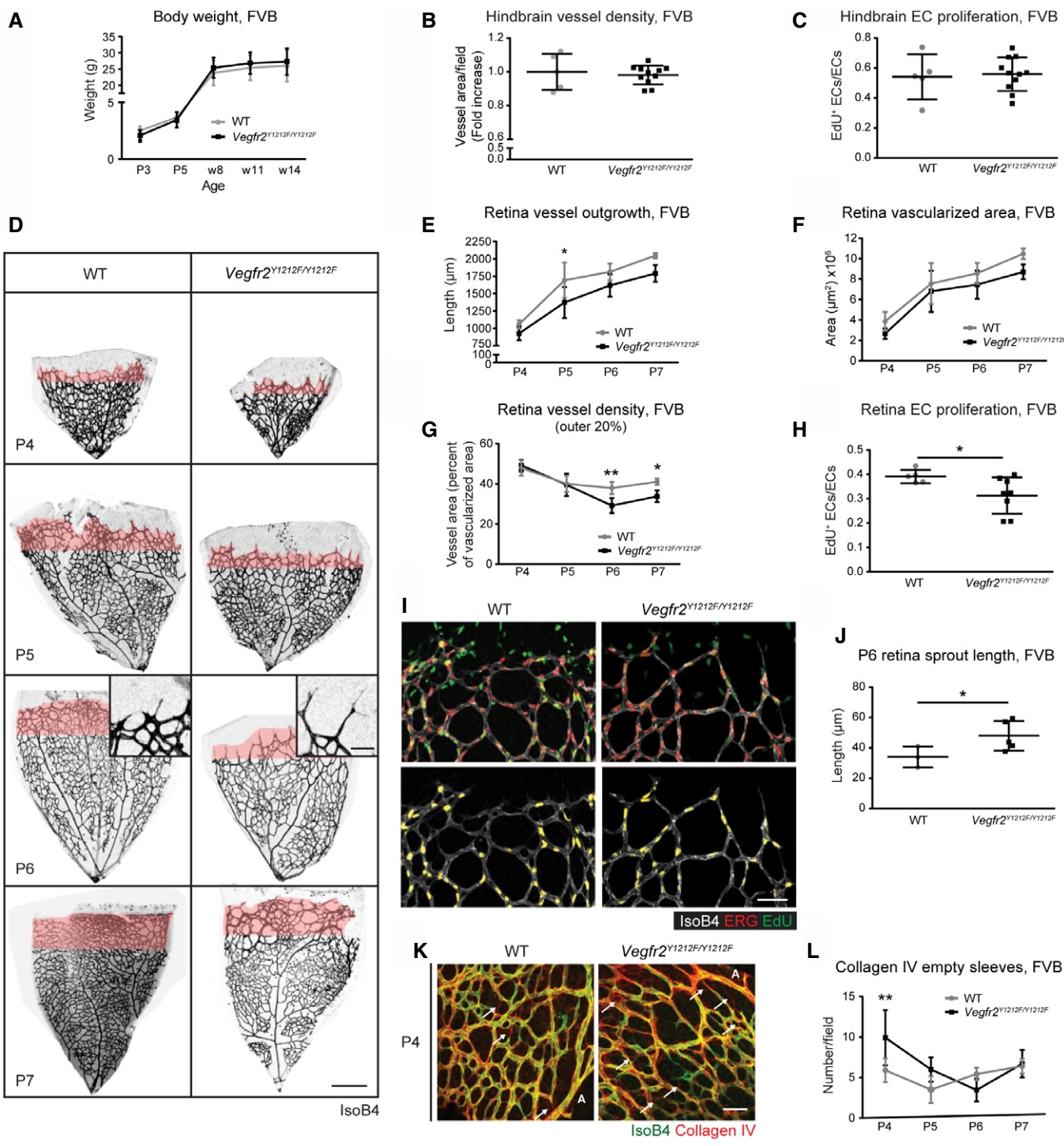

**Figure 2.**

PI3Kp85, pAkt (Fig 3H and J, and EV5B), was increased in VEGFA-treated WT, but not in mutant *Vegfr2*$^{Y1212F/Y1212F}$ FVB iECs. VEGFA treatment for different time periods of WT and mutant FVB iECs followed by immunoblotting for pERK1/2 (Fig 3K and L) and pAkt (Fig 3K and M), verified suppressed signaling downstream of GRB2 and PI3Kp85 in the *Vegfr2*$^{Y1212F/Y1212F}$ iECs compared to WT.

## VEGFR2 pY1212 regulates Myc, and Myc overexpression rescues *Vegfr2*$^{Y1212F/Y1212F}$ sprouting

ERK1/2 and Akt pathways mediate positive and negative regulation of c-Myc expression [20,21]. ERK1/2 phosphorylates c-Myc on S62, preventing its proteasomal degradation [20], while Akt phosphorylates several proteins negatively regulating c-Myc nuclear

translocation, including Foxo1 [20,33]. c-Myc protein levels were examined by immunoblotting of lung lysates from FVB and C57Bl/6 mice that received VEGFA through tail-vein injection. WT, but not *Vegfr2^{Y1212F/Y1212F}*, mice showed very rapid VEGFA-induced increase in c-Myc levels in FVB (Fig 4A and B). Comparing c-Myc

protein levels between the mouse strains using freshly isolated, non-cultured iECs showed higher protein levels in the C57Bl/6 strain than the FVB strain, irrespective of the Y1212F VEGFR2 knock-in (Fig 4C and D). In agreement, *c-Myc* transcript levels were higher in C57Bl/6 than in FVB iECs irrespective of the

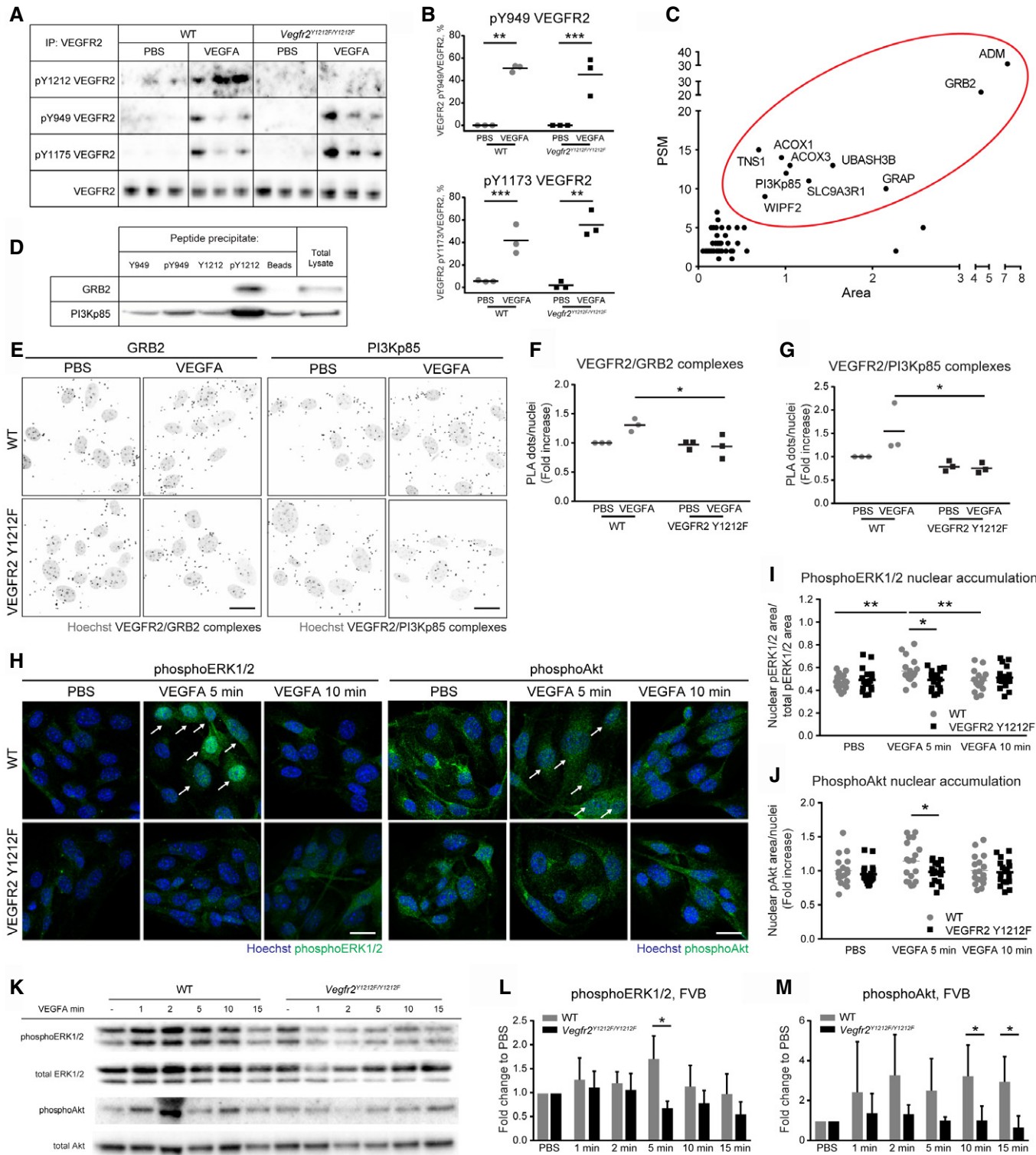

**Figure 3.**

**Figure 3. Specific binding of VEGFR2 Y1212 to GRB2 and PI3Kp85.**

A, B   Immunoblot showing VEGFR2 phosphorylation on pY1212, pY949, and pY1173. (A) Anti-VEGFR2 immunoprecipitates (IP) from lung lysates from wild-type (WT) and *Vegfr2*$^{Y1212F/Y1212F}$ FVB mice, tail-vein-injected with PBS or VEGF for 1 min. (B) Quantification as percentage of phosphoprotein level normalized to total VEGFR2. Each lane (A) and each dot (B) represent one individual mouse. Sidak's multiple comparison test, *P < 0.05, **P < 0.01, ***P < 0.001. n = 3.

C   Summary plot of 51 proteins enriched in the immobilized pY1212 peptide precipitate. Ten proteins (red oval) were selected for further analysis. PSM: peptide-spectrum match. Area: average area of spectra for the three most abundant peptides in arbitrary units.

D   Blot for peptide-enriched GRB2 and PI3Kp85. HDMEC total lysate incubated with indicated immobilized peptides followed by immunoblotting for GRB2 and PI3Kp85.

E–G   VEGFR2/GRB2 and VEGFR2/PI3Kp85 complexes in isolated (i) ECs. (E) PLA for VEGFR2/GRB2 and VEGFR2/PI3Kp85 complexes (black puncta) in iECs from FVB WT and *Vegfr2*$^{Y1212F/Y1212F}$ lungs. Hoechst 33342 show nuclei (gray). Scale bar, 20 μm. (F, G) Fold increase over PBS-treated sample; each dot represents the mean of six fields. Two-way ANOVA P = 0.0206 (F), P = 0.0368 (G); Sidak's multiple comparison test, *P < 0.05. n = 3 experiments.

H   phosphoERK1/2 and phosphoAkt immunostaining (green) of iECs from FVB WT and *Vegfr2*$^{Y1212F/Y1212F}$ lungs. Hoechst 33342 (blue) shows nuclei. Arrows indicate nuclear accumulation of phosphoERK1/2 and phosphoAkt, respectively. Scale bar, 20 μm.

I   Nuclear accumulation expressed as phosphoERK1/2-positive nuclear area normalized to the total phosphoERK1/2 area by field. Each dot represents 1 field. Min, minutes of stimulation. Two-way ANOVA P = 0.0248; Sidak's multiple comparison test, *P < 0.05, **P < 0.01. n = 3 experiments.

J   Nuclear accumulation expressed as fold increase of nuclear phosphoAkt area in *Vegfr2*$^{Y1212F/Y1212F}$ over PBS-treated WT iECs, normalized to the number of nuclei. Each dot represents 1 field. Min, minutes of stimulation. Unpaired *t*-test, *P < 0.05. n = 3 experiments.

K   Immunoblot for phosphoERK1/2, total ERK1/2, phosphoAkt, and total Akt on lung lysates from WT and *Vegfr2*$^{Y1212F/Y1212F}$ FVB mice tail-vein-injected with PBS or VEGFA followed by circulation for time points indicated. Each lane represents one individual mouse.

L   Quantification of phosphoERK1/2/total ERK1/2 expressed as fold change to PBS. Min, minutes of stimulation. Error bars: SD; 2-way ANOVA P = 0.0025; unpaired *t*-test, *P < 0.05. n = 3 experiments.

M   Quantification of phosphoAkt/total Akt expressed as fold change to PBS. Min, minutes of stimulation. Error bars: SD; 2-way ANOVA P = 0.0003; unpaired *t*-test, *P < 0.05. n = 3 experiments.

Source data are available online for this figure.

mutation (Fig 4E). VEGFA-induced proliferation of C57Bl/6 as well as FVB iECs was higher in the WT than in the *Vegfr2*$^{Y1212F/Y1212F}$ cells, judged from the extent of EdU incorporation (Fig 4F–I). *c-Myc* silencing suppressed EdU incorporation into WT C57Bl/6 and FVB iECs to a similar extent, while incorporation into the Y1212F mutant iECs was comparable in control and *c-Myc*-silenced conditions. These data indicate that VEGFA-induced proliferation of WT iECs occurred in a c-Myc- and pY1212-dependent manner in both C57Bl/6 and FVB strains.

We confirmed the role of c-Myc downstream of pY1212 in VEGFR2, by examining offspring from a cross between *Vegfr2*$^{Y1212F/Y1212F}$ C57Bl/6 and inducible Cdh5-CreERT2: R26StopFLMYC C57Bl/6 (MycOE/Cre$^+$) mice [34,35] where tamoxifen treatment of the latter strain results in the excision of a floxed STOP cassette and the subsequent EC-specific overexpression of CAG promoter-driven

human MYC gene (MycOE). Increased transcription level of the human MYC gene in E11.5 MycOE Cre$^+$ embryos, treated with tamoxifen from E9.5, was confirmed by qPCR (Fig EV5C). Still, the EC proliferation defect in the *Vegfr2*$^{Y1212F/Y1212F}$ hindbrains (see Fig 1D) could not be verified at E11.5, in part due to loss of embryos after tamoxifen treatment, but possibly also due to insufficient duration of MYC overexpression following tamoxifen administration. We therefore examined angiogenic sprouting and EC proliferation in WT and mutant C57Bl/6 E11.5 embryo explants with and without MycOE, cultured in collagen gels for 10 days. Angiogenic sprouting was low in *Vegfr2*$^{Y1212F/Y1212F}$ C57Bl/6 MycOE/Cre$^-$ embryos compared with WT Cre$^+$ and Cre$^-$ littermates (Fig 4J and K). Importantly, *Vegfr2*$^{Y1212F/Y1212F}$ MycOE/Cre$^+$ C57Bl/6 explants produced angiogenic sprouts to a significantly higher extent compared to MycOE/Cre$^-$ littermates. Thus, the angiogenic

**Figure 4. VEGFR2 pY1212 regulates Myc and rescue of *Vegfr2*$^{Y1212F/Y1212F}$ proliferation defect by Myc overexpression.**

A   Immunoblot for Myc on lung lysates from wild-type (WT) and *Vegfr2*$^{Y1212F/Y1212F}$ FVB mice tail-vein-injected with PBS or VEGFA followed by circulation for time points indicated. Each lane represents one individual mouse.

B   Quantification of Myc/actin expressed as fold change to PBS. Min, minutes of stimulation. Error bars: SD; 2-way ANOVA P = 0.0058; unpaired *t*-test, *P < 0.05, **P < 0.01. n = 3 experiments.

C, D   Myc protein expression. (C) Myc and actin immunoblotting on lysates of lung isolated ECs from untreated WT and mutant (VEGFR2 Y1212F) C57Bl/6 and FVB mice. (D) Quantification of Myc/actin expressed as arbitrary units. Each lane (C) and each dot (D) represent one individual mouse. Two-way ANOVA P < 0.0001; unpaired *t*-test, **P < 0.01. n = 5–7.

E   Baseline transcriptional level of Myc from lung isolated ECs from untreated WT and mutant (VEGFR2 Y1212F) C57Bl/6 and FVB mice. Each dot represents one individual mouse. Two-way ANOVA P = 0.0041; Sidak's multiple comparison test, *P < 0.05. n = 4–5.

F–I   Proliferation of C57Bl/6 and FVB iECs. (F, G) CD31 (white), ERG (red), EdU (green), and double-positive cells (yellow) in C57Bl/6 (F) and FVB (G) iECs. EdU signal was masked with ERG signal. (H, I) EdU incorporation in ERG-positive ECs normalized to total ERG-positive ECs. Each dot represents 2 fields/well, one well/mouse. Scale bar, 100 μm. Error bars: SD; 2-way ANOVA P = 0.0843 (H), P = 0.0474 (I); unpaired *t*-test, *P < 0.05; **P < 0.01. C57Bl/6 n = 3–6 mice, FVB n = 3–5 mice.

J, K   Rescue of angiogenic sprouting by Myc overexpression in E11.5 C57Bl/6 *Vegfr2*$^{Y1212F/Y1212F}$ explant tissue. (J) Representative images of E11.5 explants from C57Bl/6 embryos taken from WT and *Vegfr2*$^{Y1212F/Y1212F}$ littermates with (MycOE/Cre$^+$) and without (MycOE/Cre$^-$) endothelial-specific overexpression of Myc. Endothelial cell marker (CD31) immunostaining highlights the vasculature in white. Embryonic tissue blacked out to highlight sprouting area. (K) Quantification as fold increase in vessel sprouting area compared to WT MycOE/Cre$^-$. Scale bar, 250 μm. Error bars: SD; unpaired *t*-test, **P < 0.01, ****P < 0.0001. n = 4–11 explants, one explant/embryo.

L, M   Rescue of EC proliferation. (L) Myc overexpression in E11.5 C57Bl/6 *Vegfr2*$^{Y1212F/Y1212F}$ explant tissue. ERG (red), EdU (green), and double-positive cells (yellow; arrows). (M) Quantification of EdU/ERG double-positive ECs normalized to total ECs. Scale bar, 50 μm. Error bars: SD; unpaired *t*-test, ***P < 0.001, ****P < 0.0001. n = 3–7 explants, one explant/embryo.

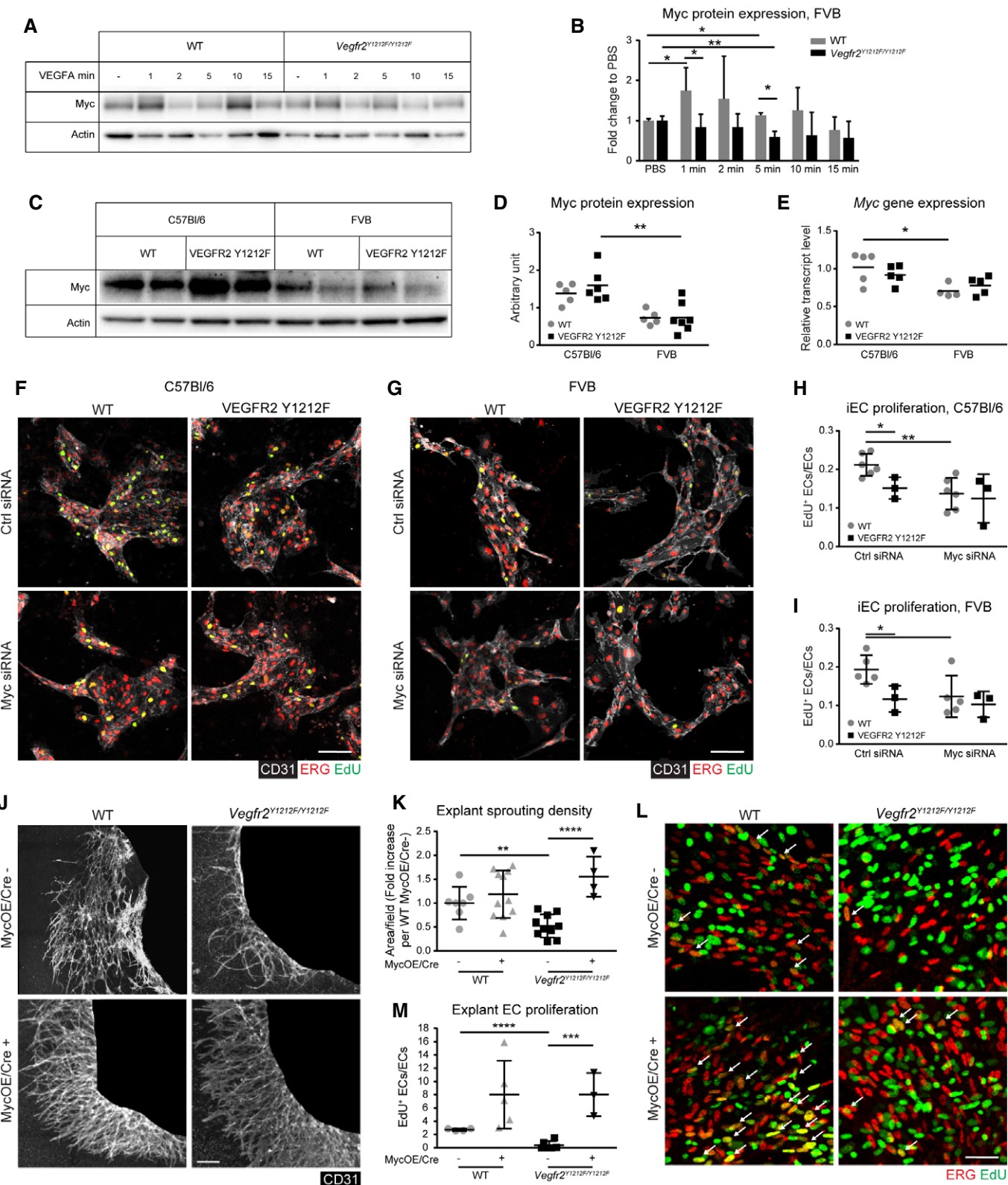

**Figure 4.**

sprouting defect in the C57Bl/6 mutants was rescued by Myc over-expression. Proliferation was assessed by quantification of EC-specific EdU incorporation (Fig 4L and M), which showed a similar MycOE-dependent increase in WT and *Vegfr2*$^{Y1212F/Y1212F}$ EC proliferation. We conclude that overexpression of human Myc rescued the proliferation-deficient Y1212F phenotype in the explant assay.

### c-Myc-dependent transcriptional activity is regulated by VEGFR2 Y1212 signaling

Gene regulation was next examined in WT and VEGFR2 Y1212F iECs treated with VEGFA for 1 or 3 h. GSEA on transcripts arrays showed both loss and gain of gene regulation in the VEGFR2 Y1212F condition. Gene sets for which there was a significant lack of upregulation in the VEGFR2 Y1212F vs. WT-derived iECs are listed in Table 2. Gene sets "K-Ras signaling up" (transcriptional regulators) and "Angiogenesis" (endothelial regulators) were downregulated in all VEGFR2 Y1212F iEC conditions (C57Bl/6 and FVB; basal and

VEGFA-treated; Fig 5A and B). MTORC1 (mammalian target of rapamycin complex 1) activated downstream of PI3K/Akt is critical in promoting anabolism and angiogenesis and in suppressing autophagy [36]. The MTORC1 gene set was upregulated in the 3-h VEGFA-treated WT C57Bl/6 iECs, but not in VEGFR2 Y1212F iECs (Fig 5A), in agreement with the loss of the PI3K/Akt pathway in the mutant iECs.

However, GSEA did not detect significant regulation in typical Myc-curated gene sets, which could depend on the *in vitro* culture condition. Therefore, regulation of a selected set of core Myc-dependent genes was analyzed by qPCR on lung tissue from mice, tail-vein-injected or not with VEGFA (Fig 5C–F). Analysis of

**Table 2.  GSEA gene sets enriched in wild-type (WT) vs. VEGFR2 Y1212F ECs.**

| | C57Bl/6 | | | FVB | | |
|---|---|---|---|---|---|---|
| | WT vs. VEGFR2 Y1212F[a] | ES/NES | FDR q | WT vs. VEGFR2 Y1212F[b] | ES/NES | FDR q |
| Gene sets Basal condition | EMT | 0.54/2.22 | 0.0000 | EMT | 0.49/2.07 | 0.0000 |
| | Myogenesis | 0.49/2.04 | 0.0005 | UV response dn | 0.49/1.95 | 0.0005 |
| | Angiogenesis | 0.59/1.81 | 0.0047 | Wnt, bcatenin | 0.53/1.79 | 0.0033 |
| | K-Ras signaling up | 0.40/1.67 | 0.0134 | *Hedgehog signaling* | 0.57/1.79 | 0.0025 |
| | UV response dn | 0.39/1.55 | 0.0333 | K-Ras signaling up | 0.39/1.69 | 0.0046 |
| | Hypoxia | 0.38/1.54 | 0.0302 | Angiogenesis | 0.45/1.43 | 0.0357 |
| | Wnt, bcatenin | 0.47/1.53 | 0.0300 | TGFbeta signaling | 0.39/1.39 | 0.0446 |
| | Adipogenesis | 0.37/1.52 | 0.0303 | *Myogenesis* | 0.32/1.37 | 0.0449 |
| Gene sets 1-h VEGFA | EMT | 0.51/2.10 | 0.0000 | EMT | 0.54/2.25 | 0.0000 |
| | Myogenesis | 0.47/1.96 | 0.0014 | *UV response dn* | 0.50/2.01 | 0.0000 |
| | Angiogenesis | 0.60/1.79 | 0.0047 | *WNT, bcatenin* | 0.55/1.81 | 0.0008 |
| | Coagulation | 0.42/1.67 | 0.0113 | K-Ras signaling up | 0.43/1.77 | 0.0015 |
| | K-Ras signaling up | 0.39/1.63 | 0.0143 | *Hypoxia* | 0.41/1.70 | 0.0029 |
| | K-Ras signaling dn | 0.37/1.50 | 0.0403 | *Apoptosis* | 0.41/1.66 | 0.0050 |
| | | | | Myogenesis | 0.38/1.55 | 0.0131 |
| | | | | *Hedgehog signaling* | 0.48/1.52 | 0.0145 |
| | | | | *TGFbeta signaling* | 0.44/1.52 | 0.0132 |
| | | | | Angiogenesis | 0.48/1.48 | 0.0182 |
| | | | | *TNFa signaling* | 0.34/1.43 | 0.0262 |
| | | | | *P53 pathway* | 0.34/1.43 | 0.0246 |
| Gene sets 3-h VEGFA | EMT | 0.55/2.24 | 0.0000 | EMT | 0.51/2.14 | 0.0000 |
| | Myogenesis | 0.47/1.91 | 0.0014 | *WNT, bcatenin* | 0.57/1.82 | 0.0023 |
| | Adipogenesis | 0.46/1.85 | 0.0027 | UV response down | 0.45/1.80 | 0.0018 |
| | UV response down | 0.46/1.78 | 0.0043 | K-Ras signaling up | 0.43/1.77 | 0.0017 |
| | Angiogenesis | 0.58/1.78 | 0.0040 | Angiogenesis | 0.53/1.67 | 0.0069 |
| | Hypoxia | 0.42/1.71 | 0.0082 | *Hedgehog signaling* | 0.49/1.54 | 0.0139 |
| | K-Ras signaling up | 0.42/1.69 | 0.0085 | Myogenesis | 0.37/1.54 | 0.0125 |
| | Glycolysis | 0.38/1.55 | 0.0031 | Hypoxia | 0.37/1.52 | 0.0130 |
| | Coagulation | 0.40/1.53 | 0.0032 | *Apoptosis* | 0.36/1.44 | 0.0230 |
| | Apical junction | 0.37/1.50 | 0.0040 | | | |
| | MTORC1 signaling | 0.37/1.48 | 0.0436 | | | |

[a]Gene sets showing gene upregulation in WT ECs compared to VEGFR2 Y1212F ECs isolated from mouse lungs and left untreated (Basal) or treated with 50 ng/ml VEGFA for 1 or 3 h.
[b]Gene sets in italics indicate unique upregulation in the FVB vs. C57Bl/6 strains, when comparing similar conditions (i.e., basal compared to basal, VEGFA for 1 h with VEGFA for 1 h).

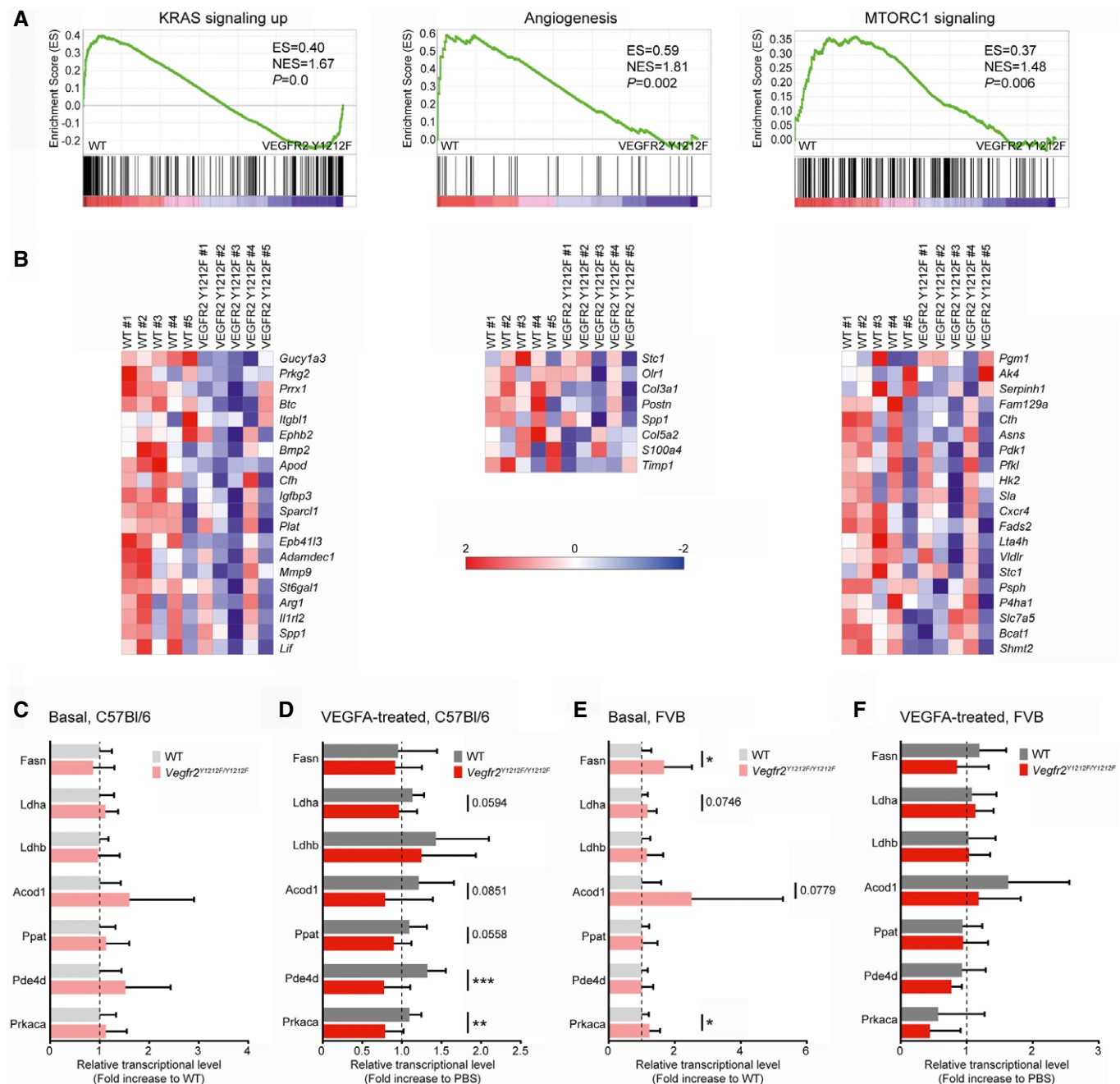

**Figure 5. Myc gene transcription and transcriptional activity regulation by VEGFR2 Y1212 signaling.**

A  GSEAs showing upregulation of genes belonging to the K-Ras, angiogenesis, and mTORC1 gene sets in wild-type (WT) compared to VEGFR2 Y1212F isolated ECs. K-Ras and angiogenesis gene sets derived from C57Bl/6 iECs kept in basal condition; mTORC1 gene set derived from C57Bl/6 iECs treated with VEGFA for 3 h. See Table 2. Enrichment score (ES) and normalized ES (NES) are shown. *P*-values < 0.05 were regarded as significant.

B  Heat maps displaying gene regulation in gene sets shown in (A). The color in the heatmap shows the gene expression value, expressed as a z-score across animals. Red color represents relative higher expression, and blue color represents relative lower expression value. The z-score is calculated by $(x–x')/s$ where $x$ is individual gene expression, $x'$ is the mean of the gene expression across samples, and $s$ is standard deviation of the gene expression across samples.

C–F  Myc-regulated gene expression downstream of VEGFR2 pY1212. Quantitative PCR analysis of a set of known Myc-regulated transcripts in C57Bl/6 (C, D) and FVB (E, F) adult lungs at basal (C, E) and at 1 h after tail-vein administration of VEGFA (D, F). FASN, fatty acid synthase; Ldha and Ldhb, lactate dehydrogenase A and B; Acod1, aconitate decarboxylase 1; Ppat, amidophosphoribosyltransferase; Pde4d, cAMP-specific 3′,5′-cyclic phosphodiesterase 4D; Prkaca, cAMP-dependent protein kinase catalytic subunit alpha. Error bars: SD; 2-way ANOVA between WT and *Vegfr2*[Y1212F/Y1212F] $P = 0.0453$ (C), $P < 0.0001$ (D), $P = 0.0018$ (E), $P = 0.09$ (F); unpaired *t*-test, *$P < 0.05$, **$P < 0.01$, ***$P < 0.001$. C57Bl/6 $n = 8$–11 mice, FVB $n = 9$–14 mice.

transcripts isolated after VEGFA/PBS circulation for 1 h showed significant VEGFA-induced upregulation of selected Myc-dependent genes in WT but not *Vegfr2*^*Y1212F/Y1212F*^ C57Bl/6 mice (Fig 5C and D). In contrast, VEGFA-regulated expression was, overall, not observed in the FVB strain, neither in WT nor in *Vegfr2*^*Y1212F/Y1212F*^ mice (Fig 5F). At baseline, the *Vegfr2*^*Y1212F/Y1212F*^ FVB-derived samples displayed significantly higher expression of c-Myc-regulated genes compared to the WT FVB samples (Fig 5E). We hypothesize that the higher basal level of c-Myc-dependent gene expression in the FVB mutant mice may be a result of compensation. Overall, these data demonstrate a strain-specific pattern of dysregulation of key c-Myc-regulated genes in the VEGFR2 Y1212F mutant.

## Discussion

We show here that the Y1212 phosphosite in VEGFR2 is critical for initiation of GRB2/ERK1/2 and PI3K/Akt pathways. This signaling is crucial for c-Myc induction and EC proliferation. The importance of pY1212 signaling in EC function is demonstrated by that 50% of the mutant *Vegfr2*^*Y1212F/Y1212F*^ progeny succumbed during C57Bl/6 embryogenesis. In contrast, FVB mutant mice survived embryogenesis but displayed disturbed postnatal vascular growth and patterning in the retina and trachea. Postnatal baseline c-Myc protein levels and transcripts were higher in C57Bl/6 than in the FVB. Whether the different baseline c-Myc levels contributed to the different VEGFA sensitivities in regulation of a core of c-Myc-dependent genes (sensitive and Y1212-dependent in the C57Bl/6 but insensitive in the FVB) is unclear at this point. We suggest that compensatory mechanisms bypassed the Y1212F mutant deficiency in the FVB and in the surviving C57Bl/6 embryos. It is well established that different mouse strains display different angiogenic phenotypes likely due to single nucleotide polymorphisms (SNPs) in key regulators [37–39]. In this context, we note that the size of the proliferating EC pool differed between the strains. In the WT C57Bl/6 hindbrain, 32.4% ± 13 SD of the ECs incorporated EdU, while in the WT FVB hindbrain, 54.1% ± 15 SD of all ECs showed EdU incorporation. In the postnatal retina, 39% ± 3 SD of the FVB but 50.7% ± 9 SD of the C57Bl/6 EC pool showed EdU incorporation. This pattern may reflect the presence of strain-specific SNPs controlling basal EC proliferative capacity under different developmental phases. We conclude that vascular beds with the lower extent of EdU incorporation in the WT condition appeared more vulnerable in the mutant setting, which may have contributed to the establishment of the different strain-specific *Vegfr2*^*Y1212F/Y1212F*^ phenotypes.

Both C57Bl/6 and FVB-derived *Vegfr2*^*Y1212F/Y1212F*^ iECs showed deficient gene regulation in the basal condition that agreed with Y1212-dependent activation of GRB2/ERK1/2 pathway such as the gene set "K-Ras signaling up". VEGFA-induced gene regulation in the gene set "MTORC1 signaling", in accordance with deficient PI3K/Akt signaling, was established only in the C57Bl/6 and not in the FVB background. The very broad effect of the Y1212F mutation on gene regulation is in agreement with c-Myc pathway involvement but does not exclude the engagement of other transcriptional effectors. The role of c-Myc downstream of VEGFR2 pY1212 was, however, emphasized by the induction of c-Myc protein in WT but not Y1212F mutant iECs, reduced EdU incorporation in *c-Myc*-silenced WT iECs to a level comparable to that in mutant iECs, as

well as the rescue of the sprouting defect and impaired EC proliferation by overexpression of human MYC in *Vegfr2*^*Y1212F/Y1212F*^ C57Bl/6 embryo explants.

Global *c-Myc* gene deletion leads to defects in vasculogenesis and primitive erythropoiesis, in part due to loss of c-Myc-regulated VEGF expression [40]. Combined endothelial and hematopoietic cell-specific loss of *c-Myc*, through excision by constitutive *Tie2-Cre*, induces similar vascular defect as global loss of *c-Myc*, with complete lethality beyond E12.5 [41]. Upon Tie1-regulated Cre excision of the *c-Myc* gene, which favors endothelial rather than hematopoietic excision, embryos perish later (E17.5) and half survive to birth [41]. The phenotype of these *c-Myc*-deficient mouse strains of mixed background, combined with the fact that aortic endothelial cells in *Tie2-Cre:c-Myc*^−/−^ mice at E10.5 incorporated EdU to the same extent as the Myc expressing control, led He *et al* to suggest that c-Myc is dispensable for EC proliferation. We show here the role of the genetic background for the establishment of a strain-specific Myc-dependent phenotype. Importantly, blood cell parameters were unaffected by the *Vegfr2*^*Y1212F/Y1212F*^ mutation in adult FVB and C57Bl/6 mice (Table 1). Postnatal conditional deletion of *c-Myc* in ECs using *Pdgfb*-creERT2 on the C57Bl/6 background gives rise to delayed retinal vascular outgrowth with a thinned and poorly branched vasculature at P5, thus, in overall agreement with the phenotype of the retinal vasculature in *Vegfr2*^*Y1212F/Y1212F*^ FVB pups [33]. Moreover, EC-specific Myc overexpression in mice causes edema and hemorrhaging, leading to embryonic lethality [42]. Thus, both gain and loss of Myc function are deleterious, in keeping with an essential role for Myc in cellular homeostasis.

Numerous signaling pathways have been implicated in VEGFA-induced EC proliferation, typically involving ERK1/2 [3], but differing in the mode of pathway initiation. RTK-induced activation of ERK1/2 is as a rule initiated by the binding of GRB2s SH2 domain to a pY-X-N sequence [43] such as pY(775) and D (aspartic acid) - N (asparagine) in the PDGF receptor-β [44]. In agreement, the Y1212 residue in VEGFR2 is followed by D - N. VEGFR2 proliferative signaling can be also induced in a Ras-independent fashion via binding of PLCγ to VEGFR2 pY1173, leading to PLCγ activation and ERK1/2 activation via protein kinase C [10,45]. The relative contribution of the two phosphorylation sites, pY1212 described here and pY1173 [10,45] in VEGFA/VEGFR-dependent proliferation, remains to be determined. The evidence for an essential role for pY1212 in ERK1/2 pathway induction includes: (i) the loss of VEGFR2/GRB2 complex formation in the Y1212F VEGFR2 mutant, (ii) reduced pERK1/2 induction and nuclear translocation in response to VEGFA, and (iii) reduced EdU incorporation in VEGFA-treated VEGFR2 Y1212F iECs compared to WT. Moreover, (iv) GSEA analyses showed loss of K-Ras gene regulation in VEGFR2 Y1212F iECs.

Although VEGFR2 is known to induce the PI3K/Akt pathway, it has remained unclear which phosphorylation site of VEGFR2 is involved in the binding of PI3Kp85. One identified pathway involves VEGFR2 pY949 signaling leading to downstream c-Src activation and phosphorylation of the RTK Axl which in turn mediates PI3K/Akt activation [46]. Our data indicate that PI3Kp85 binds to pY1212, leading to suppressed Akt signaling in the Y1212F mutant VEGFR2. The evidence for a role for pY1212 in induction of PI3K/Akt signaling includes: (i) loss of PI3Kp85/VEGFR2 complex formation in the Y1212F iECs, (ii) reduced pAkt induction and nuclear translocation

in the VEGFA-treated mutant Y1212F iECs, and (iii) loss of the Akt pathway-dependent GSEA gene set MTORC1 in the VEGFA-treated VEGFR2 Y1212F C57 iECs. The loss-of-proliferation phenotype of *Vegfr2*[Y1212F/Y1212F] agrees with the decreased EC proliferation seen upon postnatal deletion of *Akt1*, one of the two isoforms expressed in ECs [47]. However, the net outcome of c-Myc regulation resulting in EC proliferation depends on a balance between c-Myc expression levels and Akt-mediated phosphorylation of FOXO1, limiting c-Myc nuclear translocation [33,48]. In the primary iECs analyzed here, accumulation of pThr24 FOXO1/3, known to correlate with FOXO1-dependent nuclear exclusion of c-Myc, was not detected in the basal condition, nor after treatment with VEGFA. VEGFA-induced pThr24 has been reported before, but is considerably augmented under conditions of exaggerated pAkt accumulation such as upon inhibition of the pAkt regulator phosphatase and tensin homolog, PTEN [49,50]. Still, analyses of EdU incorporation in WT and *Vegfr2*[Y1212F/Y1212F] iECs provided proof of *c-Myc*-dependent loss in proliferative capacity in the mutant iECs, indicating that suppressed pERK1/2 signaling dominated in the mutant setting. On the other hand, it is conceivable that the loss in postnatal vessel stability manifested in the FVB retina and trachea was dependent on reduced PI3K/Akt signaling [51,52].

In summary, our data emphasize the importance of examining signal transduction in relevant *in vivo* models. We conclude that VEGFR2 pY1212 is critical in EC proliferation and vessel stability, operating via the activation of GRB2/ERK1/2 and PI3K/Akt pathways controlling c-Myc gene regulatory activity in strain-specific patterns.

# Materials and Methods

### Animal studies

Animal work was approved by the Uppsala University board of animal experimentation (permit 5.2.18-8927-16). The ethical application procedure conforms with the "Animal Research: Reporting of *In Vivo* Experiments (ARRIVE)" guidelines [53]. All strains were maintained by heterozygous breeding, and analyses were performed on littermates from these breedings. Mice were as a rule kept in groups of max 5/cage. Plastic cages had floors covered with a layer of wood shavings and contained enrichment. Cages, food, and water bottles were changed once/week, and supervision was carried out daily by trained staff. Constitutive mixed background knock-in mice with the tyrosine 1212 residue of VEGFR2 changed to phenylalanine [15] were backcrossed separately to the C57Bl/6 and FVB (The Jackson Laboratory, 000664 and 001800, respectively) backgrounds for more than 10 generations. C57Bl/6 embryos were collected at E11.5 and E14.5. Retinas were collected and investigated from postnatal day (P) 4 to P7, while tracheas were harvested and investigated at P1, P3, and P5. C57Bl/6: R26StopFLMYC allele mice were purchased from The Jackson Laboratory (020458) [54]. C57Bl/6: Cdh5-CreER mice were kindly provided by R. Adams [Max Planck Institute Münster] [34]. No gender-specific effects of the mutation were observed. All animal experiments were repeated at least three independent times with WT and mutant mice compared within the same litter. Sample size was chosen to ensure reproducibility and allow stringent statistical analysis. Randomization of mice and blinding of

the investigators were not performed. No mice/data points were excluded from analyses.

### EC proliferation

EC proliferation in yolk sac, hindbrain, retina, and iECs was examined using Alexa Fluor 488 and 647 Click-iT EdU Imaging Kits (Invitrogen). 5-ethynyl-2′-deoxyuridine (EdU) was prepared at 10 mg/ml in 10% dimethylsulfoxide (DMSO) for pregnant females, 5 mg/ml in 10% DMSO for pups, and for iECs, 10 μM in MV2 medium (PromoCell) with supplements (5% FCS, 5 ng/ml hEGF, 0.5 ng/ml VEGF, 20 ng/ml R3 IGF, 1 μg/μl ascorbic acid, 10 ng/ml bFGF, and 0.2 μg/μl hydrocortisone; PromoCell). Pregnant females received 200 μl EdU intraperitoneally (ip) 1 h before sacrifice, while 5 μl/gram body weight was injected ip into P6 and P7 pups 4 h before sacrifice. Embryos were collected and fixed overnight in 4% paraformaldehyde (PFA). Yolk sacs were collected and fixed for 2 h in 4% PFA. Eyes were enucleated and fixed in 4% PFA for 30 min at room temperature. iECs were incubated with EdU for 5 h and then fixed in 4% PFA. Hindbrains and retinas were dissected and immunostained to detect ERG and isolectin B4 (IsoIB4). Yolk sacs and iECs were immunostained with ERG and CD31 antibodies. Thereafter, samples were incubated for EdU detection.

### siRNA transfection

siRNA transfection was done by using Lipofectamine RNAiMax (Thermo Fisher Scientific). Control or Myc siRNA (OriGene) was added to the cells at a concentration of 20 nM together with the transfection reagent. The cells were changed to fresh medium next day, and the experiments were done 48 h after transfection.

### Isolation of ECs from mouse lung

Mouse lungs were harvested between P8 and P10, minced, and digested in 10 ml of 2 mg/ml collagenase type I (385 U/mg; Worthington) in D-phosphate-buffered saline (PBS) with $Ca^{2+}$/ $Mg^{2+}$ for 1 h at 37°C. Digested lungs were filtered through a 70-μm disposable cell strainer (BD Falcon) and centrifuged at 400 *g* for 8 min at 4°C. Cell pellets were suspended in 0.1% bovine serum albumin (BSA) in DPBS without $Ca^{2+}$/$Mg^{2+}$ and incubated with rat anti-CD31 (MEC 13.3, BD Pharmingen, 553370; RRID:AB_394816)-conjugated magnetic beads (sheep anti-rat IgG Dynabeads, Invitrogen) for 30 min at room temperature. Beads were separated with a magnet and washed six times with DPBS without $Ca^{2+}$/$Mg^{2+}$. iECs bound to the beads were suspended in Dulbecco's modified Eagle's medium (DMEM) GlutaMAX™ growth medium (Gibco) supplemented with 10% FCS, 50 μg/ml heparin, 1% penicillin/strepto-mycin (Sigma), and 1% non-essential amino acids (Gibco), and seeded in 1% gelatin-coated 12-well plates. After 6 days, cells were incubated with trypsin (Gibco), collected with a magnet, and seeded in 8-well chamber slides. Following incubation in DMEM Gluta-MAX™ with 0.5% FCS for 2 h, the iECs were treated with 50 ng/ml mouse VEGFA165 (VEGFA, PeproTech). Thereafter, the slides were placed on ice, washed in cold PBS with 1 mM $Na_3VO_4$, and fixed in 4% PFA for 15 min followed by immunostaining and/or proximity ligation assay (PLA).

Mouse lung ECs were also isolated using a slightly different procedure using MACS cell isolation equipment and reagents (Miltenyi Biotec) and following their instructions. Briefly, mouse lungs were dissociated by using Mouse Lung Dissociation Kit (Miltenyi Biotec). The endothelial cell population was labeled by incubation with CD31 MicroBeads (Miltenyi Biotec) and separated on MACS magnetic separator (Miltenyi Biotec). The cells from each mouse were either used directly or split into two wells of 12-well plate and cultured in MV2 medium with supplements (PromoCell). The unattached cells were removed next day, and fresh medium was changed every day to maintain the cells until they were used for immunoblotting, transcript analysis, proliferation assay, and siRNA experiments.

## Peptide design

Synthetic peptides of 18–19 residues, phosphorylated or not, corresponding to the Y949 and Y1212 regions in human VEGFR2, were designed with a biotin modification at the N-terminus to allow immobilization onto streptavidin-coated beads. Peptides were dissolved according to their isoelectric point: Y949 in $dH_2O$, pY949 in 10% acetic acid/4%TFA/$dH_2O$, and Y1212 and pY1212 in 0.016 mM $NH_4OH$/$dH_2O$. Peptides used in the process were as follows:

hVEGFR2 Y949: GARFRQGKDYVGAIPVDL
hVEGFR2 pY949: GARFRQGKDpYVGAIPVDL
hVEGFR2 Y1212: EVCDPKFHYDNTAGISQYL
hVEGFR2 pY1212: EVCDPKFHpYDNTAGISQYL

## Validation of the Y1212F mutation and peripheral blood parameters

The nucleotide sequence of $Vegfr2^{Y1212F/Y1212F}$ was confirmed by Sanger sequencing at the Uppsala Genome Center core facility (SciLifeLab.se). A roughly 700 bp section of the mouse $Vegfr2$ gene, spanning exon 27 and parts of the surrounding introns, was amplified (Fig EV1A) and agarose gel extracted (Qiagen, QIAquick Gel Extraction Kit). The sequence files (Fig EV1B) were assembled using ChromasPro software (Technelysium, South Brisbane, Australia) into a single contig, and an NCBI BLAST analysis [55] was used to confirm the mutation (TAT to TTC) and to identify an 86-base pair insertion in introns 26–27, remaining from the neomycin resistance cassette.

Primers used in the process were as follows:

Fwd Amplification primer—5′ AAGTAGTATGGGGACTGCCTAACC
Rev Amplification primer—5′ TAATCCTGCTCTGTGTTGCTTTCC
Fwd I sequencing primer—5′ TGCACCCCCTGCCTTGGAGG
Fwd II sequencing primer—5′ TTGAAAGGGATGTGTGGAAGAAGG
Rev I sequencing primer—5′ TCCTCTTCACTGCTTTCTGATTCTCC
Rev II sequencing primer—5′ AGCACACACAGTCAAAGACACC

For peripheral blood cell count on young adult mice, 1 ml blood was collected in Eppendorf tubes with $K_3$EDTA at a final concentration of 10 mM and immediately used for platelet (PLT), white blood cell (WBC), and red blood cell (RBC) counting using the Sysmex XP-300 Automated Hematology Analyzer (Sysmex Corporation, Kobe, Japan).

## Immunofluorescent staining

Whole-mount staining of hindbrains, retinas, and tracheas and immunostaining of iECs were performed as follows. Hindbrains were fixed with the embryo (full body fixation), dissected, and incubated with blocking solution [1% BSA/0.25% Triton X-100 in phosphate-buffered saline (PBS)]. Samples were immunostained with primary and secondary antibodies overnight at 4°C and then washed with PBlec (1 mM $CaCl_2$/1 mM $MgCl_2$/0.1 mM $MnCl_2$/1% Triton X-100 in PBS), before the overnight incubation at 4°C with biotin-conjugated isolectin B4 (Sigma L2140; RRID:AB_2313663).

Retinas dissected from enucleated eyes fixed in 4% PFA were incubated in 1% BSA/2% FCS/0.05% Na-deoxycholate/0.5% Triton X-100/0.02% Na azide in PBS. After incubation with primary antibodies and secondary antibodies, retinas were equilibrated in PBlec and stained with biotin-conjugated isolectin B4 (Sigma L2140) overnight at 4°C and subsequently with streptavidin (Molecular Probes).

Tracheas were dissected and fixed in cold methanol, blocked in 0.3% Triton X-100/3% BSA in PBS, and incubated with primary and secondary antibodies and Hoechst 33342.

iECs were incubated for 1 h in 3% BSA/5% FCS/0.2% Tween-20 in PBS, with primary/secondary antibodies and Hoechst 33342.

All samples were mounted using Fluoromount-G (SouthernBiotech). Microscopy was done with a Zeiss LSM 700 Microscope with AxioCam HRm and Zen software (Zeiss), and/or Leica TCS SP8 Confocal Microscope with PMT and HyD detectors and LAS X Navigator software version 3.5.2.18963 (Leica). Image acquisition was done with 10×, 20×, and 63× objectives for both retina and trachea tissues. Processing and quantification of images were done with ImageJ software (NIH) or CellProfiler (Broad Institute) [56] to minimize the effect of subject bias. Quantification was normalized to the total vascularized area (the area of the retina extending from the optic nerve to the vascular front) or to the vessel area (isolectin B4-positive area in retina or CD31-positive area in trachea) or to the number of ERG-positive ECs or to the total number of nuclei.

## In situ proximity ligation (PLA)

Cells were stimulated with murine $VEGFA_{165}$ (PeproTech; denoted VEGFA throughout) for 5 min, washed with cold PBS, and fixed with 4% PFA as described above. Protein/protein complexes were visualized by in situ PLA using the Duolink® Detection Kit (Sigma) with Duolink® blocking solution and detection protocol. PLA negative control conditions were as follows: only secondary antibodies conjugated with oligonucleotides and only single primary antibodies (anti-VEGFR2, anti-GRB2, or anti-PI3Kp85) combined with both secondary antibodies conjugated with oligonucleotides. VEGFR2-VEGFR2 PLA complexes using secondary antibodies conjugated with two different oligonucleotides were used as positive controls. Staining of nuclei with Hoechst 33342 was done after "detection" during the last wash steps. The number of PLA signals was quantified using CellProfiler (Broad Institute) [56] and normalized to the number of nuclei.

## Antibodies

Biotin-conjugated isolectin B4 (Sigma, L-2140) and hamster anti-CD31 antibody (clone 2H8, Thermo Scientific, MA3105; RRID:AB_223592)

were used for endothelial cell immunostaining in retina and trachea, respectively. Rabbit anti-ERG (clone EPR3864, Abcam, ab92513; RRID:AB_2630401) was used in hindbrain and retina to identify EC nuclei. Goat anti-collagen IV antibody (Millipore, AB769; RRID:AB_92262) was used to demonstrate the presence of empty basement membrane ghosts (empty sleeves). Rabbit anti-Golph4 antibody (Abcam, ab28049; RRID:AB_732692) was used to detect the EC polarity. Apoptosis was investigated using a rabbit anti-cleaved caspase-3 antibody (Cell Signaling, #9661; RRID:AB_2341188). Goat anti-podocalyxin (R&D, AF1556; RRID:AB_354858) was used to detect proper lumen formation in trachea. iECs were stained with rabbit anti-phospho (p)ERK1/2 (Thr202/Tyr204) (clone 197G2, Cell Signaling, #4377; RRID:AB_331775) and rabbit anti-pAkt (Ser473) (Cell Signaling, #9271; RRID:AB_329825). Goat anti-mouse VEGFR2 (R&D, AF644; RRID:AB_355500), rabbit anti-GRB2 (clone C-23, Santa Cruz, sc-255; RRID:AB_631602), and rabbit anti-PI3Kp85 (Millipore, 06-195; RRID:AB_310069) were used to detect PLA complex formation in iECs. Embryo explants were stained with a goat anti-CD31 antibody (R&D system, AF3628; RRID:AB_2161028).

Western blotting membranes were incubated with rabbit anti-GRB2 (clone C-23, Santa Cruz, sc-255; RRID:AB_631602), rabbit anti-PI3Kp85 (Millipore, 06-195; RRID:AB_310069), rabbit anti-pVEGFR2 (Tyr1212) (clone 11A3, Cell Signaling, #2477; RRID:AB_331374), rabbit anti-pVEGFR2 (Tyr951) (Cell Signaling, #2471; RRID:AB_331021), rabbit anti-pVEGFR2 (Tyr1175) (clone 19A10, Cell Signaling, #2478; RRID:AB_331377), rabbit anti-VEGFR2 (clone 55B11, Cell Signaling, #2479; RRID:AB_2212507), mouse anti-Myc (clone 9E10, Santa Cruz, sc-40; RRID:AB_627268), rabbit anti-pAkt (Thr308) (Cell Signaling, #9275; RRID:AB_329828), rabbit anti-Akt (Cell Signaling, #9272; RRID:AB_329827), rabbit anti-pErk1/Erk2 (p-p44/42 MAPK Thr202/Tyr204) (clone 197G2, Cell Signaling, #4377; RRID:AB_331775), rabbit anti-Erk1/Erk2 (p44/42 MAPK) (Cell Signaling, #9102; RRID:AB_330744), goat anti-actin (clone C-11, Santa Cruz, sc-1615; RRID:AB_630835), and mouse anti-GAPDH (clone 6C5, Millipore, MAB374; RRID:AB_2107445) as primary antibodies and with horseradish peroxidase (HRP)-conjugated secondary anti-rabbit, anti-mouse, or anti-goat (GE Healthcare) antibodies.

For fluorescence-activated cell sorting, the following antibodies were used: Fc receptor antibody (clone 93, Invitrogen, 14-0161-85; RRID:AB_467134) and conjugated antibodies FITC-CD31 (BD Pharmingen, 553372; RRID:AB_394818) and APC-CD45 (clone 30-F11, BioLegend, 103112; RRID:AB_312977).

## Migration assay

Mouse lung iECs were seeded at the density of 50,000 cells/well into a 96-well ImageLock™ tissue culture plate (Essen BioScience) and incubated in Dulbecco's modified Eagle's medium (DMEM) GlutaMAX™ growth media (Gibco) supplemented with 10% FCS, 50 μg/ml heparin, 1% penicillin/streptomycin (Sigma), and 1% non-essential amino acids (Gibco) overnight followed by standardized scratching using WoundMaker™ (Essen BioScience). Cells were washed with PBS and stimulated with 50 ng/ml mouse VEGFA (PeproTech) in DMEM GlutaMAX™ with 0.5% FCS. The plate was placed into an IncuCyte ZOOM (Essen BioScience) and scanned every 15 min for 12 h using a 10× objective during which data were collected using ZOOM software (Essen BioScience). Cellular

migration was analyzed using MTrackJ in ImageJ (NIH). Single cell tracks were selected manually, and data were collected for quantitative analysis of migration speed. Migration speed of one cell was obtained by dividing the migration distance (measured by the software) by time.

## Embryo explant assay

$Vegfr2^{Y1212F/+}$ C57Bl/6: R26Stop$^{FL}$MYC allele mice and $Vegfr2^{Y1212F/+}$ C57Bl/6: Cdh5-CreER mice were put together for timed breeding. At E9.5, 100 μl of 4OHT-tamoxifen (Sigma, H7904) (10 mg/ml dissolved in peanut oil) was injected intraperitoneally into pregnant females. At E11.5, embryos were harvested and stored on ice in PBS until prepared for collagen embedding by dissecting away the head and the tail of each embryo. Genotyping of the embryos was done using the embryonic tailpiece. The embryos were embedded between two layers of rat tail type I collagen (Life technologies) and cultured in DMEM supplemented with 15% FCS and 30 ng/ml VEGFA for 10 days. Samples were fixed in 4% PFA and analyzed by microscopy after immunostaining with antibodies. Tile scan z-stack images were taken with a Leica TCS SP8 Confocal Microscope with PMT and HyD detectors and LAS X Navigator software (Leica). Image analysis was done using ImageJ software (NIH) on a 2,000 × 2,000 μm region of the sprouting front of each explant.

## Heart rate and heart staining

Heartbeats in E11.5 embryos stored in PBS were counted manually by scoring movies rendered by collecting time-lapse images of each embryo for 1 min. Microscopy was performed using Leica M205FA Stereo Microscope (Leica).

Hematoxylin and eosin staining to visualize embryonic cardiac development was performed on 10-μm cryosections made from OCT-embedded tissues. Briefly, embryos were fixed in 4% PFA overnight and then placed in 30% sucrose at 4°C for 48 h before embedding in OCT. Transverse sections were made using a cryostat, followed by staining and imaging using a Leica DMi8 Microscope with DFC7000T camera and LAS X software (Leica).

## Fluorescence-activated cell sorting (FACS)

E11.5 embryos were minced using sharp scissors and placed in 500 μl 4 mg/ml collagenase IV (Life Technologies) and 0.2 mg/ml DNaseI (Roche), 10% FBS (Life Technologies) in PBS, and incubated for 15 min at 37°C while shaking at 600 rpm. The tissue was further disrupted by repeated pipetting using a 1-ml plastic pipette, every 5 min. Enzyme activity was quenched by adding 1 ml of FACS buffer (PBS, 0.5% FBS, 2 mM EDTA), and the solution was passed through a 70-μm nylon filter (BD biosciences). Cells were collected and resuspended in 100 μl FACS buffer containing Fc receptor antibody (Invitrogen, 14-0161-85, clone 93) for 15 min. After FcR blocking, cells were incubated with directly conjugated antibodies FITC-CD31 (BD Pharmingen, 553372, clone MEC 13.3) and APC-CD45 (BioLegend, 103112, clone 30-F11) staining for 30 min at room temperature. Cells were then washed and resuspended in FACS buffer to be read by the BD CytoFLEX flow cytometer (Beckman Coulter) at the Uppsala University BioVis platform.

## Mass spectrometry (MS) sample preparation

HDMECs (Human dermal microvascular endothelial cells; Promo-Cell) were cultured on dishes coated with 1% gelatin using endothelial cell medium MV2 (PromoCell) with all supplements at 37°C and 5% $CO_2$. Cells at four to six passages were used. Cells were neither authenticated nor tested for potential mycoplasma contamination after arrival to the laboratory.

Cell was placed on ice and washed with hypotonic lysis buffer (10 mM HEPES pH 7.9, 2 mM $MgCl_2$, 10 mM KCl, 0.1% Na-deoxycholate). Then, lysis buffer supplemented with Protease Inhibitor Cocktail (Roche) and 1 mM sodium orthovanadate was added and cells were scraped from the plate. The cell suspension was left 15 min at room temperature before drawing it, with single rapid strokes, through a syringe with narrow-gauge hypodermic needle (no. 27) five times. The suspensions were then incubated for 45 min at 4°C on a rotating wheel before repeating the mechanical lysis step 10 times. The cell lysate supernatant was transferred to a new tube after 20 min of centrifugation at 11,000 g.

The cell lysate, 2 mg in 1 ml lysis buffer, was incubated with biotinylated peptides (1 nM) immobilized on 20 μl pre-cleaned streptavidin-coated Sera-Mag SpeedBeads (GE Healthcare) for 1 h at 4°C on a rotating wheel. The beads were isolated using a magnet for 3 min and washed 3 times with lysis buffer and dried. The dried beads were then resuspended in 25 mM ammonium bicarbonate with 1 mM dithiothreitol (DTT) and incubated for 1 h at 37°C on a shaking platform. Next, iodoacetamide was added to a final concentration of 5 mM and samples were incubated 10 min in the dark. Trypsin digestion was subsequently performed overnight at 37°C after addition of DTT to a final concentration of 5 mM and trypsin to a final concentration of 0.02 μg/μl. Supernatants containing peptides were then collected and dried using a SpeedVac.

## Liquid chromatography (LC)-MS and MS analysis

For each LC-MS run, dried peptides were dissolved in mobile phase A [95% water, 5% dimethylsulfoxide (DMSO), 0.1% formic acid] and injected by the autosampler (Ultimate 3000 RSLC System, Thermo Scientific Dionex) into an Acclaim PepMap nanotrap column (C18, 3 μm, 100 Å, 75 μm × 20 mm), and subsequently separated on an Acclaim PepMap RSLC column (C18, 2 μm, 100 Å, 75 μm × 50 cm; Thermo Scientific). Peptides were separated using a gradient of mobile phase A and B (90% ACN, 5% water, 5% DMSO, 0.1% FA), ranging from 6 to 37% B in 240 min with a flow of 0.25 μl/min.

Online LC-MS was performed using a hybrid Q Exactive mass spectrometer (Thermo Scientific). FTMS master scans with 70,000 resolution (and mass range 300–1,700 m/z) were followed by data-dependent MS/MS (35,000 resolution) on the top 10 ions using higher energy collision dissociation (HCD) at 30% normalized collision energy. Precursors were isolated with a 2 m/z window. Automatic gain control (AGC) targets were 1e6 for MS1 and 1e5 for MS2. Maximum injection times were 100 ms for MS1 and 150 ms for MS2. The entire duty cycle lasted ~1.5 s. Dynamic exclusion was used with 60-s duration. Precursors with unassigned charge state or charge state 1 were excluded. An underfill ratio of 1% was used.

The MS raw files were searched using Proteome Discoverer 1.4 with SequestHT for protein identification. Precursor mass tolerance was set to 10 ppm and for fragments to 0.02 Da. Oxidized methionine was set as dynamic modification and carbamidomethylation as static modification. The algorithm considered tryptic peptides with a maximum of two missed cleavages. Spectra were matched to human subset of the UniProt database (including isoforms, downloaded on June 11, 2015), and results were filtered to 1% false discovery rates (FDR).

The MS analyses revealed retention of approximately 3,000 proteins to the different peptide matrices. Proteins detected in all conditions (binding equally well to immobilized Y1212, pY1212, Y949, and pY949 peptides) were excluded, reducing the list of retained proteins to 167. The peptide-spectrum match (PSM) indicates the total number of identified peptides matched for a given detected protein; a PSM of 1 means that 1 peptide was detected once for the protein of interest. A high PSM number and a high area score are both indicators of peptide abundance in the sample. Fifty-one proteins binding to pY1212 had a PSM > 1, and of these, we chose the proteins that showed the highest area score (average area of the spectra of the three most abundant peptides). This selection limited the list to 10 proteins being specifically enriched for binding to pY1212.

## Western blotting

SDS–PAGE on 4–12% Bis-Tris polyacrylamide gel (Novex by Life Technologies) followed by transfer to an Immobilon-P PVDF membrane (Millipore) and immunoblotting was performed as described in Ref. [6].

## Microarray and Gene Set enrichment analysis (GSEA)

Mouse lung ECs were isolated from WT or $Vegfr2^{Y1212F/Y1212F}$ mice (both C57Bl/6 and FVB strains) using MACS lung disassociation kit and CD31 MicroBeads (Miltenyi Biotec). The iECs were cultured individually in 6-well plates until confluence and then plated in 12-well plates for experiments. The cells from each mouse were starved for 3 h in MV2 medium with 0.2% FCS and then either left untreated (basal) or treated with VEGFA (50 ng/ml) for 1 or 3 h. RNA was isolated immediately by using RNeasy Micro Plus Kit (Qiagen). RNA concentration was measured by NanoDrop (Thermo Fisher Scientific), and RNA integrity number (RIN) was assessed by Bioanalyzer (Agilent, samples with a RIN > 7 were used for microarray). Microarray was performed by using Clariom™ D Assays, mouse (Thermo Fisher Scientific) with RNA samples from 5 WT, and 5 $Vegfr2^{Y1212F/Y1212F}$ mice for each time point. GSEA was performed with 50 hallmark gene sets from MSigDB database version 6.2 ("http://software.broadinstitute.org/gsea/msigdb"). GSEA is a computational method that determines whether an *a priori* defined set of genes shows statistically significant concordant differences between two biological states (e.g., phenotypes) [57]. Analyses were filtered to 5% FDR.

## Quantitative PCR

PBS or 250 ng/g mouse VEGFA (PeproTech) was injected into 8- to 10-week-old WT or $Vegfr2^{Y1212F/Y1212F}$ mice via the tail vein and circulated for 1 h before the mice were sacrificed. Lungs were isolated and snap-frozen immediately. RNA from the lungs was extracted and purified using an RNeasy Mini Kit (Qiagen). RNA concentrations were measured in a NanoDrop spectrophotometer

(Thermo Fisher Scientific) and adjusted to equal concentrations, followed by reverse transcription using SuperScript III (Life Technology). Real-time quantitative PCR was performed on a Bio-Rad CFX96 Real-Time PCR Machine using SsoAdvanced SYBR Green Supermix (Bio-Rad). The housekeeping gene ribosomal protein L19 (*Rpl19*) was used as an internal control. The comparative Ct method was used to calculate fold differences in gene expression. For further analysis and data visualization, basal gene expression in C57Bl/6 and FVB *Vegfr2*$^{Y1212F/Y1212F}$ samples was expressed as the fold change in gene expression as compared to the C57Bl/6 and FVB WT mean expression, respectively. Similarly, for all four groups: C57Bl/6 WT, FVB WT, C57Bl/6 *Vegfr2*$^{Y1212F/Y1212F}$, and FVB *Vegfr2*$^{Y1212F/Y1212F}$, the VEGFA-induced change in gene expression was expressed as fold change compared to the average gene expression in the corresponding PBS-treated group.

### Sample size and statistical analysis

All experiments were repeated at least three independent times (biological repeats). Sample size was chosen to ensure reproducibility and allow stringent statistical analysis. Statistical analysis was performed with Excel (Microsoft) and GraphPad Prism (GraphPad). Normality of each group was assumed before performing statistical analysis inasmuch groups are biological samples. The chi-square test was used to determine the difference between the expected Mendelian frequencies of genotypes and the observed frequencies in the *Vegfr2*$^{Y1212F/Y1212F}$ mice on the C57Bl/6 and FVB backgrounds. Different summary statistics were evaluated in hindbrains, retinas, tracheas, and iECs for each mouse. Values were compared between the two groups, WT and *Vegfr2*$^{Y1212F/Y1212F}$ or WT and VEGFR2 Y1212F using unpaired two-sample *t*-test. Two-way ANOVA with Sidak or Bonferroni correction for multiple comparisons was used to compare samples belonging to more than 2 variables and between groups with non-equal variance. All tests were two-tailed, and $P < 0.05$ was considered a statistically significant result.

## Data availability

Data and reagents are available upon reasonable request. The mass spectrometry proteomics data have been deposited to the ProteomeXchange Consortium via the PRIDE partner repository (www.ebi.uk.pride/archive/projects/PXD014517) with the dataset identifier PXD014517.

The transcript array data generated using iECs were deposited in the Gene Expression Omnibus (GEO) archive; reference no GSE136085 (www.ncbi.nlm.nih.gov/geo/query/acc.cgi?acc = GSE136085).

*Expanded View* for this article is available online.

### Acknowledgements
The authors acknowledge the equipment and expert advice supplied by the BioVis imaging core facility (Uppsala), the SciLifeLab Clinical Proteomics Mass Spectrometry facility, and the Array and Analysis Facility, Uppsala University. This study was made possible through grants to Lena Claesson-Welsh from the Swedish Research Council (2015-02375), the Swedish Cancer Foundation (CAN2016/578), and the Knut and Alice Wallenberg Foundation (KAW 2015.0030). KAW also supported LCW with a Wallenberg Scholar grant (2015.0275).

### Author contributions
The project was conceived by CT and LC-W; experiments were planned by CT, ROS, YJ, MHel, and LC-W; experiments were performed by CT, ROS, YJ, PM, MHed, and MS-J; GSEA data were analyzed by YS, ROS, YJ, and LC-W; all other data were analyzed and evaluated by CT, ROS, YJ, and LC-W; the *Vegfr2*$^{Y1212F/Y1212F}$ mice were constructed by MS; the manuscript was written by CT and LC-W with comments from all authors.

### Conflict of interest
The authors declare that they have no conflict of interest.

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
