## [Review Process File · EMBO Reports]

Myc-dependent endothelial proliferation is controlled by phosphotyrosine 1212 in VEGF receptor-2

Chiara Testini, Ross O. Smith, Yi Jin, Pernilla Martinsson, Ying Sun, Marie Hedlund, Miguel Sáinz-Jaspeado, Masabumi Shibuya, Mats Hellström, Lena Claesson-Welsh

Review timeline:

Submission date:	31 January 2019
Editorial Decision:	6 March 2019
Revision received:	11 July 2019
Editorial Decision:	9 August 2019
Revision received:	18 August 2019
Accepted:	26 August 2019

Editor: Deniz Senyilmaz-Tiebe

Transaction Report:

1st Editorial Decision

6 March 2019

Thank you for submitting your manuscript for consideration by EMBO Reports. Three referees agreed to review your manuscript. So far, we have received two referee reports that are copied below. Given that both referees are in fair agreement that you should be given a chance to revise the manuscript, I would like to ask you to begin revising your study along the lines suggested by the referees.

Please note that this is a preliminary decision made in the interest of time, and that it is subject to change should the third referee offer very strong and convincing reasons for this. As soon as we will receive the final report on your manuscript, we will forward it to you as well.

As you can see, both referees express interest in the proposed role of VEGFR2/Myc signaling in vascular development. However, they also raise concerns that need to be addressed in full before we can consider publication of the manuscript here.

Given these constructive comments, I would like to invite you to revise your manuscript with the understanding that the referee must be fully addressed and their suggestions taken on board. Please address all referee concerns in a complete point-by-point response. Acceptance of the manuscript will depend on a positive outcome of a second round of review. It is EMBO Reports policy to allow a single round of revision only and acceptance or rejection of the manuscript will therefore depend on the completeness of your responses included in the next, final version of the manuscript.

REFeree REPORTS

Referee #1:

In this paper the authors analyze *in vivo* the role of axis VEGFR2/Myc in vascular development by exploiting a knock-in mouse model expressing VEGFR2 Y1212F in C57Bl/6 and FVB mice. They demonstrate that the substitution of Y1212 with F impairs the correct activation GRB2 and PI3Kp85 resulting in reduction of endothelial proliferation and impairment of vascular development. Interestingly the phenotype differs in C57Bl/6 and FVB background. Actually the former shows embryonic defects, while the latter only impairment in post-natal vascularization (retina and trachea).

This is an interesting paper with a solid analysis of biochemical mechanisms and mice analysis. However, the connection between the phenotype difference between the two KI respectively in Bl6 and FVB mice has to be carefully better analyzed

Fig 1 and 2. Proliferation analysis has to be evaluated *in vitro* in EC isolated from lung of KI mice generated in the 2 strains

Fig 3. Besides PLA, the complex formation between VEGFR2-Y1212 has to be validated by pull-down and co-IP assays in cells carrying different constructs: VEGFR2, VEGFR2-Y1212F and the appropriate controls. Similarly the nuclear accumulation of pErk e and pAKT has to be confirmed by blot analysis of subcellular fractions.

This referee requires some information why the authors focused their investigative efforts on Myc. Furthermore the section "Myc-dependent transcriptional activity is regulated by VEGFR2 Y1212 signaling" has to be re-considered by a careful comparison both *in vitro* and *in vivo* of the role of Myc in the two strains. Which is the Myc related gene profiling in ECs (not only in whole lung) isolated from the two KI mice with different background? Does Myc overexpression modify the response (proliferation, migration) of EC isolated from Bl6 VEGFR2Y1212F mice? How do the author prove that the increased amount of Myc in FVB is related to post-transcriptional events? Does the reduction of Myc expression (shRNA) in FVB-derived ECs rescue the Bl6 phenotype (migration, proliferation)? Which are the differences between the *in vitro* and *in vivo* transcriptomes?

Referee #2:

Testini et al have provide new insights into how different phospho-tyrosine residues in the VEGFR2, the major VEGFA signalling receptor, contribute to signal transduction by the receptor. The authors make several novel findings in this study. First, they demonstrate that pY1212 has an essential role in signal transduction by VEGFR2 in vascular development *in vivo*. This was demonstrated by generating site-specific Y1212F mutant mice. Previous studies mutating this site did not find it had an essential role but as Testini et al show, the phenotype is dependent on genetic background, potentially explaining why such a role was not found previously. Second, they demonstrate that pY1212 is essential for recruiting GRB2 and PI3Kp85 to VEGFR2, potentiating ERK1/2 and PI3K signalling. This adds new knowledge for how ERK1/2 and PI3K can be activated by VEGFR2. Thirdly, the authors link defective ERK1/2 & PI3K signalling in Y1212F mutants to impaired Myc activation. Myc is known to promote endothelial cell proliferation and the authors showed that restoring Myc expression in the Y1212F mutants rescued the endothelial cell proliferation defect. Finally, VEGFR2 pY1212 has previously been reported as a binding site for Nck1/2 and the establishment of polarity necessary for endothelial cell migration. Testini et al did not find evidence for Nck1/2 binding to pY1212. Y1212F mutants showed no defect in flow-induced endothelial polarization *in vivo* and no migratory defect in culture. This argues in favor of a proliferative rather than migratory defect being responsible for the reduced vessel density in Y1212F mutants.

While the main conclusions are compelling and support the main conclusions, there are a few points

that in this reviewer's opinion would help strengthen them.

- 1) While the authors provide compelling evidence for reduced endothelial cell proliferation rather than cell migration as the cause of reduced vasculature in Y1212F mutants, is tip cell activity reduced in mutants and if so, might this at least contribute to the overall phenotype?
- 2) FoxO1 opposes Myc in cell proliferation and is repressed by PI3K/AKT. FoxO1 has an important role in regulating EC proliferation and vessel density during angiogenesis. Was FoxO1 phosphorylation or subcellular localization altered in the Y1212F mutant cells?
- 3) The conclusion that Myc levels are higher in FVB mice, explaining their lack of embryonic lethality is based on total lung lysates. How can the authors be certain that the difference in Myc signal is due to differences in endothelium and not due to contribution from other non-endothelial cell types?
- 4) Can the authors provide explanation for why FVB mice have a postnatal angiogenesis defect yet no embryonic angiogenesis defect?
- 5) In my opinion, the representative images shown for WT VEGFR2 explants +/- Myc-OE in Figure 5a show a difference in vessel sprouting similar to the difference shown by Y1212F explants +/- Myc-OE. This difference in the WT VEGFR2 representative images is not reflected in the corresponding quantitative data presented in Figure 5b. This needs to be explained and addressed.
- 6) Does Myc over-expression rescue the hindbrain vessel density defect at E11.5 if induced during embryogenesis, for example at E9.5?

Minor comments:

- 1) Dead or dying BL/6 background embryos were presumably recovered when determining mutant frequencies. Did these display signs such as hemorrhage or oedema that might offer further insight into vascular defects causing the Y1212F mutant lethality?
- 2) In addition to subcellular localisation (Fig 3 h-k), were levels of pERK1/2 and pAKT relative to total ERK1/2 or total AKT altered in the Y1212F mutants as further confirmation that these signal transduction pathways were affected?

1st Revision - authors' response

11 July 2019

Reviewer #1:

In this paper the authors analyze in vivo the role of axis VEGFR2/Myc in vascular development by exploiting a knock-in mouse model expressing VEGFR2 Y1212F in C57Bl/6 and FVB mice. They demonstrate that the substitution of Y1212 with F impairs the correct activation GRB2 and PI3Kp85 resulting in reduction of endothelial proliferation and impairment of vascular development. Interestingly the phenotype differs in C57Bl/6 and FVB background. Actually the former shows embryonic defects, while the latter only impairment in post-natal vascularization (retina and trachea).

This is an interesting paper with a solid analysis of biochemical mechanisms and mice analysis. However, the connection between the phenotype difference between the two KI respectively in Bl6 and FVB mice has to be carefully better analyzed

R1.1. *Fig 1 and 2. Proliferation analysis has to be evaluated in vitro in EC isolated from lung of KI mice generated in the 2 strains.*

Response: Figure 5 in the originally submitted manuscript, current Figure 4, panels j-m, shows an embryo explant analysis where angiogenic sprouts in the Y1212F mutant tissue display lower EdU incorporation. As a consequence, angiogenic sprouting is reduced in the mutant. Overexpression of human MYC brings up the degree of EdU incorporation and angiogenic sprouting to the same level in the WT and Y1212F mutant. This model addresses the point raised by the reviewer.

We have now complemented the explant results of Figure 4j-m, with EdU-incorporation assays using isolated endothelial cells (iECs) transfected or not with control and *Myc* siRNA, shown in Figure 4f-i. Here, we show that EdU incorporation is higher in WT iECs than that in

Vegfr2Y1212F/Y1212F iECs, both for C57Bl/6 and FVB strains. Silencing of endogenous c-Myc brings EdU incorporation down in the WT iECs, to the same level as in the *Vegfr2Y1212F/Y1212F* iECs. The pattern is the same in the two strains. These data show that EdU incorporation and therefore proliferation, in response to VEGFA occurs in a pY1212 and c-Myc-dependent manner.

R1.2. *Fig 3. Besides PLA, the complex formation between VEGFR2-Y1212 has to be validated by pull-down and co-IP assays in cells carrying different constructs: VEGFR2, VEGFR2-Y1212F and the appropriate controls. Similarly the nuclear accumulation of pErk e and pAKT has to be confirmed by blot analysis of subcellular fractions.*

Response: The results and conclusions in our study rests on analysis using iECs or using lung lysates from mice injected with VEGF. We regard this approach as a major strength of the paper. Respectfully, we ask the reviewer to note that the previous publication using a transfected cell line showed that the WT VEGFR2 but not the Y1212F mutant when overexpressed in PAE cells, formed complex with Nck [1]. Moreover, the Y1212F VEGFR2 expressing cells showed impaired migration efficiency towards VEGF. Binding of Nck to pY1212 could not be confirmed by us using capture to phosphorylated peptides followed by mass spectrometry, nor was Nck coimmunoprecipitated with VEGFR2 using lung lysates from VEGF-injected mice. Furthermore, there is no migration phenotype of Y1212F iECs (see Figure EV4g) and there is no EC polarity defect *in vitro* or *in vivo* in the mutant condition (Fig. EV4c-f), which would be expected if Nck was involved in signaling downstream of pY1212 [2]. We conclude that the previously published *in vitro* data were not reproduced in the more stringent models employed here (iECs or lung lysates from VEGFA-injected mice). In conclusion, we would prefer to avoid working with transfected, stable cell lines.

To study the interaction between VEGFR2 and GRB2/p85, we used proximity ligation assays (PLA) which entails using oligonucleotide-ligated secondary antibodies reacting with primary antibodies (in this case against p85PI3K or GRB2 on the one hand and VEGFR2 on the other) [3]. By bridging the oligos with a probe, a rolling circle amplification is initiated when the antibodies are in close proximity, i.e. when VEGFR2 has formed a complex with p85 or GRB2 (see Fig. 3f-g). The PLA results demonstrate VEGFA-induced increase in proximity between p85/GRB2 and VEGFR2. There are also PLA complexes in the absence of exogenous VEGFA, which possibly could depend on endogenous production of VEGFA by the iECs. This was not explored further; however, the PLA reactions seen in the absence of exogenous VEGFA was not a technical issue as the antibodies were titrated to create no “background”. See Figure R1. We now give details in Methods concerning the technical PLA controls, see page 21.

In contrast, we do not see VEGFA-regulated complexes in co-immunoprecipitations using lung lysates (see Figure R2) or using VEGFR2 overexpressing mouse aortic endothelial cells (data not shown) using conditions which previously allowed identification of VEGFA-induced complex formation e.g. between VEGFR2 and VE-cadherin [4]. While there was no complex formed neither before nor after VEGF-stimulation between VEGFR2 and GRB2 (data now shown), we observed a constitutive complex between VEGFR2 and p85 (Figure R2). These negative data are in accordance with the lack of previous *in vitro*-based reports on VEGF-induced complex formation between VEGFR2 and p85 or GRB2 (if it was “easy” to capture, it would have been published by now). The underlying reason is most likely that the affinities between VEGFR2 and the substrates p85 and GRB2, are relatively low.

Figure R1. PLA controls samples. a. Positive control: VEGFR2-VEGFR2 complexes detected using two secondary antibodies both recognizing the primary VEGFR2 antibody but that are conjugated to two different oligonucleotides. b-e. Negative controls. b. Only secondary antibodies. c. Only the anti-VEGFR2 antibody and both secondary antibodies. d. Only the anti-GRB2 antibody and both secondary antibodies. e. Only the anti-PI3Kp85 antibody and both secondary antibodies.

Figure R2. Co-immunoprecipitation using mouse lung lysate shows constitutive, VEGFA independent complex formation between VEGFR2 and PI3Kp85. In this analysis GRB2 migration overlaps with that of Ig light chain.

SH2 domains often show low, μM affinities towards phosphotyrosine binding (see [5], which does not preclude relevant and biologically important consequences of such low-affinity interactions. To directly measure the affinity between VEGFR2 pY1212F and the SH2 domains of p85 and GRB2 would settle this question, but it is outside the scope of this study. There is however, no doubt that VEGFA-induced signal transduction in p85 (Akt) and GRB2 (ERK1/2) dependent pathways are suppressed in the Y1212F VEGFR2 mutant expressing cells. See novel results on immunoblotting for pErk1/2 and pAkt in WT and Y1212F mutant lung lysates from VEGFAinjected mice (Figure 3k-m).

SH2 domains are known to interact with a primary sequence consisting of the phosphotyrosine and a limited number of amino acids C-terminal to the phosphotyrosine. The precipitation of

GRB2 and p85 from VEGF-stimulated endothelial cells using synthetic peptides is therefore also informative and relevant (Figure 3d). Using synthetic peptide libraries, Huang et al., [5] showed the importance of an N in position 2 C-terminal of the phosphotyrosine for binding GRB2 SH2. Kessels et al., [6] also stresses the critical importance of N in position 2 for binding of GRB2 SH2 to phosphotyrosine. Indeed, the 2nd residue C-terminal of Y1212 is N. Furthermore, GRB2 binds to a motif in PDGFRb similar to that surrounding pY1212 in VEGFR2 [7]. The rules for binding of phosphotyrosine-sequences to the SH2 domain of p85 are less strict but can also include N in position 2 [5]. As already brought up in the discussion, we do not exclude indirect activation of the PI3K/Akt pathway in response to VEGF. Thus, p85-mediated PI3K activation has been shown to also be mediated via VEGFR2-Src-dependent phosphorylation of Axl, creating a binding site for p85 [8]. See discussion, page 13, 3rd paragraph.

The nuclear accumulation of pErk and pAkt demonstrated using immunostaining (Figure 3h-j) is now complemented with immunoblotting of pErk/Erk and pAkt/Akt levels (Figure 3 k-m). By these experiments, it is evident that these VEGFR2 downstream pathways are altered in the mutant mice.

R1.3.

i) *This referee requires some information why the authors focused their investigative efforts on Myc.*

Response: On page 8 in the results, we give the background for the focus on Myc: “ERK1/2 and Akt pathways, implicated in binding to pY1212, mediate positive and negative regulation of Myc expression [9,10]. ERK1/2 phosphorylates Myc on S62, preventing its proteasomal degradation [9], while Akt phosphorylates several proteins negatively regulating Myc nuclear translocation, including FoxO1 [9,11].”

In order to introduce Myc to the general reader, we now describe the role of GRB2-Erk and p85-Akt signaling in regulation of Myc expression already in the introduction, page 4.

ii) *Furthermore the section "Myc-dependent transcriptional activity is regulated by VEGFR2 Y1212 signaling" has to be re-considered by a careful comparison both in vitro and in vivo of the role of Myc in the two strains. Which is the Myc related gene profiling in ECs (not only in whole lung) isolated from the two KI mice with different background?*

Response: To study Myc-dependent transcriptional activity, we used isolated ECs treated or not with VEGFA (current Figure 5a-b). Gene set enrichment analyses (GSEA) showed altered gene regulation in the “KRAS”, “MTORC1” and “Angiogenesis”, in agreement with loss of Grb2/p85 signaling.

In Figure 5c-d, we analyze lung lysate from VEGFA-injected mice. We consider the comparison between the VEGF-injected and non-injected samples as reflecting events in endothelial cells although we agree there may be caveats. The results are described as follows: “Analysis of transcripts isolated after VEGFA/PBS circulation for 1 h, showed significant VEGFA-induced upregulation in WT but not *Vegfr2Y1212F/Y1212F* C57Bl/6 mice (Figure 5c,d). In contrast, VEGFA-regulated

expression was, overall, not observed in the FVB strain, neither in WT nor in *Vegfr2Y1212F/Y1212F* mice (Figure 5e,f). At baseline, the *Vegfr2Y1212F/Y1212F* FVB-derived samples

displayed significantly higher expression of c-Myc-regulated genes compared to the WT FVB samples (Figure 5e,f). We hypothesize that the higher basal level of c-Myc-dependent gene expression in the FVB mutant mice may be a result of compensation. Overall, these data demonstrate a strain-specific pattern of dysregulation of key c-Myc-regulated genes in the VEGFR2 Y1212F mutant.”

We also attempted to inject VEGF intraperitoneally into P10 pups followed by isolation of endothelial cells, in order to be able to capture gene regulatory events induced by VEGFA in the mouse, i.e. true *in vivo* gene regulation, as requested. However, this approach turned out to be technically infeasible. When harvesting lungs, followed by EC isolation at 1 h after VEGF injection, we noted a VEGFA-induced decrease or no effect, combined with very spread values, in several known Myc-regulated genes. See Figure R3. It may be that we could tease out an effect using other concentrations of VEGFA, other time periods of circulation, isolate ECs from other organs or from other ages of mice etc, which we however were not able to do within the

time frame given for the revision.

Figure R3. Regulation of *Myc* pathway genes in lungs after intraperitoneal injection of VEGFA (2µg) and 1h circulation followed by harvest, RNA isolation and qPCR.

Instead we refer to the gene regulation and annotation data obtained from isolated EC (not whole lung) shown in Figure 5a,b, as well as the data from analysis on lung lysates ± VEGFA (and therefore preferentially scoring effects dependent on VEGFR2 expressed on endothelial cells) shown in Figure 5c-f.

iii) Does *Myc* overexpression modify the response (proliferation, migration) of EC isolated from *Bl6 VEGFR2Y1212F* mice?

Response: As was already shown in former Figure 5, current Figure 4j-m, human MYC overexpression induces increased EdU incorporation of ECs and sprouting angiogenesis in explants. In contrast, in iECs, there is no difference in the migratory capacity between WT and Y1212F iECs (already shown in Figure EV4g). Migration of iECs overexpressing Myc was therefore not pursued.

iv) How do the author prove that the increased amount of *Myc* in FVB is related to posttranscriptional events?

Response: Following the advice by Referee 2 (see response R2.3), we now investigated basal c-Myc protein and transcript levels in iECs. We conclude that there are no differences in c-Myc protein/transcript levels between mutant and wildtype VEGFR2 for each strain, however, protein levels are lower in the FVB than in the C57 strain. See Results, page 9, first paragraph and Discussion, page 11, first paragraph.

v) Does the reduction of *Myc* expression (*shRNA*) in FVB-derived ECs rescue the *Bl6* phenotype (migration, proliferation)?

Response: Silencing of *c-Myc* in iECs reduces EdU incorporation in the WT but not mutant iECs of both strains, see Figure 4f-i. Migration was not affected when comparing WT and Y1212F iECs (Figure EV4g) and was therefore not pursued.

vi) Which are the differences between the *in vitro* and *in vivo* transcriptomes?

Response: We refer the reviewer to the response above, R1.3ii, where we describe the different approaches we have taken to study the VEGFA-regulated *in vitro* and *in vivo* transcriptomes downstream of WT and Y1212F VEGFR2. The data are shown in Figure 5.

Reviewer #2:

Testini et al have provide new insights into how different phospho-tyrosine residues in the VEGFR2, the major VEGFA signalling receptor, contribute to signal transduction by the receptor. The authors make several novel findings in this study. First, they demonstrate that pY1212 has an essential role in signal transduction by VEGFR2 in vascular development *in vivo*. This was demonstrated by generating site-specific Y1212F mutant mice. Previous studies mutating this site did not find it had an essential role but as Testini et al show, the phenotype is dependent on genetic background, potentially explaining why such a role was not found previously. Second,

they demonstrate that pY1212 is essential for recruiting GRB2 and PI3Kp85 to VEGFR2, potentiating ERK1/2 and PI3K signalling. This adds new knowledge for how ERK1/2 and PI3K can be activated by VEGFR2. Thirdly, the authors link defective ERK1/2 & PI3K signalling in Y1212F mutants to impaired Myc activation. Myc is known to promote endothelial cell proliferation and the authors showed that restoring Myc expression in the Y1212F mutants rescued the endothelial cell proliferation defect. Finally, VEGFR2 pY1212 has previously been reported as a binding site for Nck1/2 and the establishment of polarity necessary for endothelial cell migration. Testini et al did not find evidence for Nck1/2 binding to pY1212. Y1212F mutants showed no defect in flow-induced endothelial polarization in vivo and no migratory defect in culture. This argues in favor of a proliferative rather than migratory defect being responsible for the reduced vessel density in Y1212F mutants.

While the main conclusions are compelling and support the main conclusions, there are a few points that in this reviewer's opinion would help strengthen them.

R2.1. *While the authors provide compelling evidence for reduced endothelial cell proliferation rather than cell migration as the cause of reduced vasculature in Y1212F mutants, is tip cell activity reduced in mutants and if so, might this at least contribute to the overall phenotype?*

Response: Although tip cell activity has not been studied in depth, the formation of filopodia along the sprouting front of P6 retinas was analyzed. We observed no change in the number of filopodia between the wildtype and mutant mice. See Figure EV3d,e.

R2.2. *FoxO1 opposes Myc in cell proliferation and is repressed by PI3K/AKT. FoxO1 has an important role in regulating EC proliferation and vessel density during angiogenesis. Was FoxO1 phosphorylation or subcellular localization altered in the Y1212F mutant cells?*

Response: We first attempted to study pFoxO1 accumulation using an antibody against S256 which we however later learnt from Dr. Michael Potente, is not reliable. Instead, he recommended to use a pFoxO1Thr24/FoxO3aThr32 antibody from Cell Signaling, which we bought and tested. Disappointingly, we failed to detect a signal with this antibody in the VEGFA-treated lung lysate. Therefore, we have not been able to determine whether there is a consequence of the Y1212F mutation on FoxO1 phosphorylation and function. The literature, for example see supplemental Fig. S3c in [12], shows an increase in pThr24 pFOXO1 levels in HUVECs treated with VEGFA, in parallel with accumulation of phosphoAkt although the basal level of pFOXO1 is quite high. In another paper [13], using primary mouse kidney microvascular endothelial cells, both pThr24 pFoxO1 and FoxO1 levels are very low unless Akt is overactivated using a PTEN inhibitory drug. It is possible that in primary mouse cells, pFoxO1 accumulation is harder to catch. We have also tried immunofluorescent staining however, also with technical issues. Under the given time frame, we cannot reach further on this issue. We conclude that it is possible that reduced Akt activation as a consequence of reduced p85 engagement in the Y1212F condition would further modify VEGFA-induced proliferation in the mutant, but we have not been able to settle this question. We now discuss this matter in the Discussion, page 14, first paragraph.

R2.3. *The conclusion that Myc levels are higher in FVB mice, explaining their lack of embryonic lethality is based on total lung lysates. How can the authors be certain that the difference in Myc signal is due to differences in endothelium and not due to contribution from other non-endothelial cell types?*

Response: We thank the reviewer, as well as reviewer 3, for this important comment. Using iECs we find that c-Myc levels are lower in the FVB strain (see Figure 4c-e).

We discuss the differences in the penetrance in the Y1212F mutant phenotype in the two strains in the Discussion, page 11, second paragraph: "...we note that the size of proliferating EC pool differed between the strains. In the WT C57Bl/6 hindbrain, 32.4% \pm 13 SD of the ECs incorporated EdU while in the WT FVB hindbrain, 54.1% \pm 15 SD of all ECs showed EdU incorporation. In the postnatal retina, 39% \pm 3 SD of the FVB, but 50.7% \pm 9 SD of the C57Bl/6 EC pool showed EdU incorporation. This pattern may reflect the presence of strain-specific SNPs controlling basal EC proliferative capacity under different developmental phases. We conclude that vascular beds with the lower extent of EdU incorporation in the WT condition appeared more vulnerable in the mutant setting, which may have contributed to the different, strain-specific establishment of the mutant *Vegfr2Y1212F/Y1212F* phenotypes."

R2.4. *Can the authors provide explanation for why FVB mice have a postnatal angiogenesis*

defect yet no embryonic angiogenesis defect?

Response: Postnatal angiogenesis in the eye relies on expansion of the endothelial cell pool followed by remodeling to create the mature plexus [14]. Similar expansion and remodeling are seen in the trachea (Figure EV2g-j). Although we observe reduced EC proliferation in the sprouting front, we also note a loss in vessel stability indicated by the presence of empty collagen IV sleeves, in these conditions. It is possible that other aspects of Y1212 signaling, not directly related to proliferation but rather to stability, are more dominant in the FVB strain and that it becomes established as the postnatal angiogenesis phenotype in the FVB. This is now discussed on page 14, first paragraph.

R2.5. *In my opinion, the representative images shown for WT VEGFR2 explants +/- Myc-OE in Figure 5a show a difference in vessel sprouting similar to the difference shown by Y1212F explants +/- Myc-OE. This difference in the WT VEGFR2 representative images is not reflected in the corresponding quantitative data presented in Figure 5b. This needs to be explained and addressed.*

Response: We apologize for not showing representative images. The current images now accurately reflect the observation that WT VEGFR2 explants overexpressing human Myc have a vessel sprouting similar to WT VEGFR2 explants and to Y1212F explants overexpressing Myc. The quantification relies on 4-11 observations of explants grown from C57 embryos (WTcre-7;WTcre+11;KOcre-9;KOcre+4) which we regard as highly reliable.

R2.6. *Does Myc over-expression rescue the hindbrain vessel density defect at E11.5 if induced during embryogenesis, for example at E9.5?*

Response: The intent was to study Myc overexpression in E11.5 embryos by inducing expression at E9.5 under control of the VE-cadherin/Cdh5 promoter. However, we had a number of technical issues that precluded the rescue of hindbrain vessel density/EC proliferation at E11.5. We confirmed raised Myc expression in embryo yolk sac by qPCR at E11.5 at the time of collection (Figure EV5c) but we argue that these levels were insufficient to observe an effect within the short time span of 2 days. By growing E11.5 embryo tissue in the explant experiment, we were able to extend the period in which ECs were exposed to sufficiently high levels of Myc to be able to rescue the proliferation defect. Other technical challenges may also have contributed to the lack of the expected phenotype at E11.5, for example tamoxifen toxicity (Figure R4).

Figure R4. Number of C57Bl/6 E11.5 embryos/mother with and without tamoxifen injection. T test $p < 0.05$. $n = 6-14$.

Minor comments:

R2.7. *Dead or dying BL/6 background embryos were presumably recovered when determining mutant frequencies. Did these display signs such as hemorrhage or oedema that might offer further insight into vascular defects causing the Y1212F mutant lethality?*

Response: We did not record hemorrhage or edema in the dying embryos, which quickly underwent resorption (now shown in Figure EV2a).

R2.8. *In addition to subcellular localisation (Fig 3 h-k), were levels of pERK1/2 and pAKT relative to total ERK1/2 or total AKT altered in the Y1212F mutants as further confirmation that these signal transduction pathways were affected?*

Response: We have now analyzed pErk1/2 and pAkt/Akt levels in lung lysates from VEGF-treated WT and Y1212F mice; see Figure 3k-m. In both cases do the blotting results support the engagement of the GRB2 and p85PI3K pathways downstream of pY1212 VEGFR2.

Reviewer #3

Here, Testini and colleagues address the role of Y1212 of VEGFR2 in VEGF-A initiated VEGFR2 signal transduction using a Y1212F knock-in mouse model. Of interest, they demonstrate distinct effects of this mutation on vascular development and overall survival of mutant mice that are dependent on genetic background; approximately 50% of homozygous mutant mice died between E11.5 and birth in the C57Bl/6 background, while survival of mutants in the FVB background was comparable to wild-type controls. By identifying Grb2 and PI3Kp85 as binding partners of VEGFR2 pY1212 peptide, the authors link defective induction of AKT and ERK phosphorylation downstream of VEGFR2Y1212F to reduced levels of Myc, proposed to regulate cell proliferation and metabolism important for vascular development. As the work currently stands, a number of issues need to be addressed to fully substantiate the authors' claims; these are outlined as follows:

R3.1. *Figure 1C-E: I'm not convinced that the decreases in vascular density and EC proliferation shown in the VEGFR2Y1212F/Y1212F mutant hindbrain at E11.5 would be sufficiently detrimental to result in embryonic lethality. How does vascular density appear across additional vascular beds? How do mutant embryos appear? Are there defects in cardiac or hematopoietic development that might cause/contribute to embryonic lethality?*

Response: To address this concern, we provide data on: 1) cardiac function and morphology. We observed no difference in the heart rate of WT and mutant embryos collected at E11.5. We also provide H&E staining of transverse sections through embryonic hearts showing no obvious cardiovascular developmental defects. 2) CD31 and EdU/ERG immunostaining in the yolk sac which show similar EdU incorporation in the ECs of this vasculature in WT and *Vegfr2Y1212F/Y1212F*.

3) FACS analysis of whole E11.5 embryos where endothelial cell numbers are similar between the WT and mutant. These data are now shown in Figure EV2a-h. Combined, these data support the notion that there is not one major insult that results in the embryonic death but rather an EC fragility which results in embryos that succumb during stages of development when there are e.g. metabolic challenges.

R3.2. *What underlies the decreased weight of surviving VEGFR2Y1212F/Y1212F mice in the C57Bl/6 background? Are any vascular/cardiac defects obvious? Is the decreased weight of homozygous mutants apparent past 14 weeks, ie. is the decrease transient?*

Response: Please see response above R3.1. We have performed thorough analysis of the C57 embryo and there are no major vascular/cardiac defects. Unfortunately, during the duration of the revision, we had to use a lot of mice for endothelial isolation and could not let a sufficient number of mice age past 14 weeks.

R3.3. *It is difficult to conceptualize that in a genetic background where a proportion of homozygous mutants die, there are no apparent defects in retinal angiogenesis, while in a background where all homozygous mutants survive, defects in retinal angiogenesis are present. What underlies this distinction? Is there a consequence on vascular/retinal function in surviving mice?*

Response: We suggest that the C57 embryos that survive embryogenesis are sufficiently sturdy to not show defects in postnatal development. Moreover, it is possible that the postnatal defect seen in the FVB strain involves loss of vessel stability; see above the response to R2.4. See also Discussion, page 11, second paragraph: "We suggest that compensatory mechanisms bypassed the Y1212F mutant deficiency in the FVB and in the surviving C57Bl/6 embryos. It is well established that different mouse strains display different angiogenic phenotypes likely due to single nucleotide polymorphisms (SNPs) in key regulators [15-17]. In this context, we note that the size of proliferating EC pool differed between the strains. In the WT C57Bl/6 hindbrain, 32.4%

± 13 SD of the ECs incorporated EdU while in the WT FVB hindbrain, $54.1\% \pm 15$ SD of all ECs showed EdU incorporation. In the postnatal retina, $39\% \pm 3$ SD of the FVB, but $50.7\% \pm 9$ SD of the C57Bl/6 EC pool showed EdU incorporation. This pattern may reflect the presence of strain-specific

SNPs controlling basal EC proliferative capacity under different developmental phases.

We conclude that vascular beds with the lower extent of EdU incorporation in the WT condition appeared more vulnerable in the mutant setting, which may have contributed to the different, strain-specific establishment of the mutant *Vegfr2*^{Y1212F/Y1212F} phenotypes.”

R3.4. The authors present in Figure 2k, l that *VEGFR2*^{Y1212F/Y1212F} mice have an elevated number of collagen IV positive sleeves devoid of endothelial cells at P4, but that this is not observed at P7. What factors underlie this temporal effect? If mutant mice are subjected to stress (a ROP model for example), is the vascular phenotype rescued/more pronounced?

Response: The temporal occurrence of empty sleeves due to active vessel remodeling is well documented in the literature. Please see for example the elegant paper by Franco et al. [14]. As suggested by the reviewer, we have exposed the FVB mice to a model of oxygen-induced retinopathy in order to observe the vasculature under stressed conditions. The initial stage of this retinopathy model exposes P7 pups to severe hyperoxia (75% oxygen), which promotes rapid vessel loss in both wildtype and mutant mice. In contrast to what we observed during early retina development in the Y1212F FVB mice, we saw that vessel loss in the OIR model was primarily driven by apoptosis. The resulting avascular regions created by this vessel loss are known to promote hypoxia and neovascular tuft formation, mimicking the progression of ROP. However, mice on an FVB background did not form tufts (Figure R5, panel d). Thus, the hyperoxia response in FVB mice is very different from that of C57Bl/6 mice. We show these data here (Figure R5) but do not think they are useful to show in the paper.

Figure R5. Oxygen Induced Retinopathy model leads to vessel regression by apoptosis and does not lead to neovascular tuft formation on the FVB background. a) wildtype and mutant mice exposed to 5 days of hyperoxia develop a similar avascular area; quantified in b). This vessel loss is driven by high levels of apoptosis shown in the wildtype and mutant mice by immunostaining with cleaved caspase 3 (cc3) d) OIR model is used to study neovascular growth, however FVB background mice do not develop characteristic preretinal tufts. Scale bar 1000 μ m in a) 100 μ m in c).

R3.5. In Supplementary Figure 2 g-j, the delay in sprouting capillaries, yet increase in empty collagen sleeves at P3 is confusing. Are the authors suggesting the apparent delay in sprouting is actually the consequence of increased regression?

Response: It is possible that the decreased proliferation (resulting in the sprouting delay) is not directly coregulated with the reduced stability. Both the GRB2 and p85 pathways signal in

complex patterns, not only through Myc to drive proliferation. It is possible that the postnatal FVB phenotype is more reliant on proliferation-independent signals resulting e.g in reduced stability. See page 14, first paragraph.

R3.6. *Figure 3e. Does the PLA signal detected in PBS treated Y1212F cells represent nonspecific background? Or association of VEGFR2 with GRB2/PI3Kp85 that is not VEGF-A dependent?*

Response: We do not consider the background as unspecific as we have carefully titrated the antibody levels to avoid unspecific signals. First, the best fluorescent immunostaining concentration of each antibody was tested for each antibody individually. Then, the combination of the specific antibodies with the controls was tested as shown in Figure R1 above. If the negative control of a single antibody with both secondary antibodies showed abundant PLA dots, it indicated problems with the concentration of the primaries, which then were diluted further. Under the conditions tested, the technical controls were clean. We suggest therefore that the signals seen in the absence of exogenous VEGFA could be due to endogenous production of VEGFA in the iEC cultures. Indeed, negative control tests show much fewer PLA signals than the untreated cells for the respective condition (compare Figure R1 and Figure 3e). The technical PLA controls used to optimize conditions are described in Methods, page 21.

R3.7. *The nuclear p-ERK/p-AKT data shown in Figure 3h-m is not entirely convincing. If nuclear pERK is truly as prominent as illustrated in Figure 3h, surely the signal represented in the WT panel in response to VEGF-A should be profoundly elevated compared to PBS treated cells. Have the authors assessed p-ERK/p-AKT levels by immunoblotting (this would be substantially more quantitative)? With regard to the images depicted in Figure 3h and l, the phospho-ERK and phospho-AKT channels alone should also be shown so that the reader can see where each phospho-protein is localized in the absence of confounding Hoechst signal. How does a failure to elevate ERK and AKT activity in mutant cells correlate with cellular response to VEGF-A (eg. proliferation/migration/survival)?*

Response: The complementing figure without merging with Hoechst is now shown in Figure EV 5a. We now provide blots of pERK/ERK1/2 and pAkt/Akt accumulation in response to VEGF, using lung lysates. See Figure 3k-m. In both cases do the blotting results support the engagement of the GRB2 and p85PI3K pathways downstream of pY1212 VEGFR2.

R3.8. *Based on the recent work published by Wilhelm and colleagues (Nature, 529:216-20, 2016), what happens to FOXO1 localisation in VEGFR2Y1212F/Y1212F endothelial cells following treatment with VEGF-A? Showing a difference in the proportion of nuclear/cytoplasmic FOXO1 would substantially strengthen the data suggesting that MYC levels fail to be induced in mutant cells following VEGF-A treatment (Figure 4a, b) as a result of reduced AKT activity.*

Response: Please see response to R2.2.

R3.9. *On the basis of the blot shown in Figure 4a, I'm not convinced that Myc is rapidly induced following VEGF-A treatment. The amount of protein loaded in each lane of the blots shown doesn't appear consistent across the timecourse and the transfer of protein doesn't appear consistent across the blot. Given that conclusions are being drawn on the basis of protein quantification, one should be absolutely sure that the blots are technically optimal. They could potentially be improved (and the quantification made more accurate) by loading the same amount of protein in each lane. The same applies to Figure 4c. Why is Myc apparent as different sized bands in panel C?*

Response: We agree that the quality of the Myc blots was unsatisfying. They have now been redone. Please see new Figure 4a.

R3.10. *Why are different housekeeping proteins (Actin and GAPDH) used in immunoblots depicted in Figure 4a and c?*

Response: We used actin and GAPDH with similar outcome, but show only actin in the figures. Please see Figure 4a,c.

R3. 11. *Is Figure 4c looking at Myc levels in total lung tissue? How can this be correlated with potential effects of Myc in the vasculature?*

Response: We thank the reviewer for this important comment. We have now performed analyses of Myc levels in iECs; see Figure 4c-e. Please see extensive response above to R2.3.

R3.12. *Figure 5a: why assess vascular density and proliferation in embryo explants, rather than assess the hindbrain vasculature (and other vascular beds) in MycOE mice? This really needs to be done. How was explant sprouting density calculated?*

Response: This was done, however, we believe that the MycOE mice may not express high enough levels of Myc at the early stages of hindbrain development. We also experienced problems with Tamoxifen-toxicity which caused losses of embryos, possibly with a preference for mutant embryos. Please see extensive response above to R2.6.

In the explant sprouting experiment, vascular area was calculated by selecting a standardized 2000um x 2000um window containing the region of greatest endothelial outgrowth. Vascular density within this window is reported as the CD31 positive area of vessels that extend from the implanted embryo tissue using a maximum intensity projection of this region. The embryo tissue was excluded in the calculation.

R3.13. *Can the authors detect interactions between Grb2/PI3Kp85 and VEGFR2 in coimmunoprecipitation assays?*

Response: Please, see extensive response to R1.2.

References

1. Lamalice L, Houle F, Huot J (2006) *J Biol Chem* **281**: 34009-34020
2. Dubrac A, Genet G, Ola R, *et al.* (2016) *Circulation* **133**: 409-421
3. Soderberg O, Gullberg M, Jarvius M, *et al.* (2006) *Nat Methods* **3**: 995-1000
4. Li X, Padhan N, Sjostrom EO, *et al.* (2016) *Nat Commun* **7**: 11017
5. Huang H, Li L, Wu C, *et al.* (2008) *Mol Cell Proteomics* **7**: 768-784
6. Kessels HW, Ward AC, Schumacher TN (2002) *Proc Natl Acad Sci U S A* **99**: 8524-8529
7. Yokote K, Mori S, Hansen K, *et al.* (1994) *J Biol Chem* **269**: 15337-15343
8. Ruan GX, Kazlauskas A (2012) *EMBO J* **31**: 1692-1703
9. Adhikary S, Eilers M (2005) *Nat Rev Mol Cell Biol* **6**: 635-645
10. Bouchard C, Marquardt J, Bras A, *et al.* (2004) *EMBO J* **23**: 2830-2840
11. Wilhelm K, Happel K, Eelen G, *et al.* (2016) *Nature* **529**: 216-220
12. Zhuang G, Yu K, Jiang Z, *et al.* (2013) *Sci Signal* **6**: ra25
13. Dang LTH, Aburatani T, Marsh GA, *et al.* (2017) *Biomaterials* **141**: 314-329
14. Franco CA, Jones ML, Bernabeu MO, *et al.* (2015) *PLoS Biol* **13**: e1002125
15. Nakai K, Rogers MS, Baba T, *et al.* (2009) *FASEB J* **23**: 2235-2243
16. Rohan RM, Fernandez A, Udagawa T, *et al.* (2000) *FASEB J* **14**: 871-876
17. Buyschaert I, Schmidt T, Roncal C, *et al.* (2008) *J Cell Mol Med* **12**: 2533-2551

2nd Editorial Decision

9 August 2019

Thank you for submitting the revised version of your manuscript. It has now been seen by all of the original referees.

As you can see, all referees find that the study is significantly improved during revision and recommend publication. Therefore I am pleased to accept your manuscript in principle, pending amendment of the minor/editorial issues below:

- Please address the remaining minor concerns of referee #3 textually.

REFeree REPORTS

Referee #1:

The authors performed a whole revision

Referee #2:

The authors have adequately addressed all of my questions and concerns.

Referee #3:

The authors have satisfactorily addressed the majority of the reviewers' concerns and as a result, the revised manuscript is much improved. A few minor details remain to be addressed, detailed as follows:

1. The values that the error bars represent (StdDev or SEM) should be mentioned in all figure legends.
2. I'm not sure how valid and widely accepted the measurement of heart rate *ex vivo* described by the authors is.
3. On page 8, the statement "See Methods for performed technical PLA controls" should be replaced with something like "The specificity of VEGFR2 interaction with Grb2/p85 was validated by controls omitting primary antibodies".
4. On page 8, the statement "The presence of...before VEGF-A stimulation..." should be expanded to include the possibility that the signal detected in the absence of VEGF-A treatment likely represents Grb2/p85 association with VEGFR2 that is pY1212 independent (given that this signal is also detected in Y1212F cells).
5. I remain unconvinced that the image depicting high nuclear pERK1/2 in WT EC treated with VEGF-A (Figure 3h) is representative of the majority of cells, given that the data presented in Figure 3i show a very modest, though statistically significant increase in nuclear pERK1/2 levels.
6. Page 10, the statement "Analysis of transcripts isolated after VEGFA/PBS circulation for 1 h, showed significant VEGFA- induced upregulation in WT but not Vegfr2Y1212F/Y1212F C57Bl/6 mice (Figure 5c,d)" should be revised to say selected Myc dependent genes were significantly differentially regulated, as not all genes in the panel were significantly altered in expression.

2nd Revision - authors' response

18 August 2019

Address the remaining minor concerns of referee #3 textually.

Response: This is essentially done and indicated by track changes in the uploaded manuscript. However, we have not adjusted as suggested in point 4 (On page 8, the statement "The presence of...before VEGF-A stimulation..." should be expanded to include the possibility that the signal detected in the absence of VEGF-A treatment likely represents Grb2/p85 association with VEGFR2 that is pY1212 independent (given that this signal is also detected in Y1212F cells). The reviewer is implying that VEGFR2 interactions with Grb2/p85 would be VEGFA-independent. Such speculations would, in my mind, only confuse the reader. Furthermore, points 2 and 5 where the referee expresses doubts or remains convinced, have been noted but no action has been taken.

3rd Editorial Decision

9 August 2019

Thank you for submitting your revised manuscript. I have now looked at everything and all is fine. Therefore I am very pleased to accept your manuscript for publication in EMBO Reports.

Corresponding Author Name: Lena Claesson-Welsh
Manuscript Number: EMBOR-2019-47845V1**